# The Atmospheric Bridge Communicated the $\delta^{13}C$ Decline during the Last Deglaciation to the Global Upper Ocean

**Jun Shao[1], Lowell D. Stott[1], Laurie Menviel[2], Andy Ridgwell[3], Malin Ödalen[4,5], Mayhar Mohtadi[6]**

[1]Department of Earth Science, University of Southern California, Los Angeles, CA 90089, USA

[2]Climate Change Research Centre, Earth and Sustainability Science Research Centre, University of New South Wales, NSW 2052, Sydney

[3]Department of Earth and Planetary Sciences, University of California, Riverside, CA 92521, USA

[4]Department of Meteorology, Bolin Centre for Climate Research, Stockholm University, 106 91 Stockholm, Sweden

[5]GEOMAR Helmholtz Centre for Ocean Research Kiel Duesternbrooker Weg 20 24105 Kiel, Germany

[6]MARUM-Center for Marine Environmental Sciences, University of Bremen, 28359 Germany

*Correspondence to*: Jun Shao (junshao@usc.edu)

**Abstract.** During the early part of the last glacial termination (17.2-15 ka) and coincident with a ~35ppm rise in atmospheric $CO_2$, a sharp 0.3-0.4‰ decline in atmospheric $\delta^{13}CO_2$ occurred, potentially constraining the key processes that account for the early deglacial $CO_2$ rise. A comparable $\delta^{13}C$ decline has also been documented in numerous marine proxy records from surface and thermocline-dwelling planktic foraminifera. The $\delta^{13}C$ decline recorded in planktic foraminiferal has previously been attributed to the release of respired carbon from the deep ocean that was subsequently transported within the upper ocean to sites where the signal was recorded (and then ultimately transferred to the atmosphere). Benthic $\delta^{13}C$ records from the global upper ocean, including a new record presented here from the tropical Pacific, also document this distinct early deglacial $\delta^{13}C$ decline. Here we present modeling evidence to show that rather than respired carbon from the deep ocean propagating directly to the upper ocean prior to reaching the atmosphere, the carbon would have first upwelled to the surface in the Southern Ocean where it

would enter the atmosphere. In this way the transmission of isotopically light carbon to the global upper ocean was analogous to the on-going ocean invasion of fossil fuel $CO_2$. The model results suggest that thermocline waters throughout the ocean as well as 500-2000m water depths were affected by this atmospheric bridge during the early deglaciation.

## 1. Introduction

Atmospheric $CO_2$ increased by 80-100ppm between the last glacial maximum (LGM) and the Holocene (Marcott et al., 2014; Monnin et al., 2001). During the initial ~35ppm rise in $CO_2$ between 17.2 and 15 ka, ice core records also document a 0.3‰ contemporaneous decline in atmospheric $\delta^{13}C$ (Bauska et al., 2016; Schmitt et al., 2012) (Figure 1a, b, interval highlighted in grey). Notably, this millennial-scale trend was punctuated by an interval of even more rapid change, with a 12ppm $CO_2$ increase (Marcott et al., 2014) and a -0.2‰ decrease in $\delta^{13}CO_2$ (Bauska et al., 2016) occurring in an interval of just ~200 years, between 16.3-16.1 ka (Figure 1a, b, interval highlighted in red). Hypotheses proposed to explain these observations include increased Southern Ocean ventilation (e.g. Skinner et al., 2010, Burke et al., 2012), poleward shift/enhanced Southern Hemisphere westerlies (Toggweiler et al., 2006, Anderson et al., 2009, Menviel et al., 2018) and reduced iron fertilization (Martínez-García et al., 2014, Lambert et al., 2021). However, the chain of events leading to the atmospheric changes and the location(s) where the isotope signal originated is not yet established.

Marine proxy records can provide further constraints on the possible mechanisms. For instance, during the early deglaciation, surface and thermocline dwelling foraminifera around the global ocean also recorded a distinct $\delta^{13}C$ drop (e.g. Hertzberg et al., 2016; Lund et al., 2019; Spero and Lea, 2002), an observation replicated by shallow benthic records from the tropical/subtropical

Atlantic and Indian Ocean (Lynch-Stieglitz et al., 2019; Romahn et al., 2014). These observations have been interpreted to reflect a spread of high nutrient, low $\delta^{13}C$ waters originating in the Southern Ocean that were subsequently transported throughout the upper ocean via a so-called intermediate water teleconnection (Martínez-Botí et al., 2015; Pena et al., 2013; Spero and Lea, 2002). According to this hypothesis, formerly isolated carbon from deep waters were upwelled in the Southern Ocean (Anderson et al., 2009) in response to a breakdown of deep ocean stratification (Basak et al., 2018). This carbon would have then been carried by Antarctic Intermediate Water (AAIW) and Southern Ocean Mode Water (SAMW) to low latitudes where it outgassed to the atmosphere in upwelling regions like the eastern equatorial Pacific (EEP) and recorded in ice cores. We term this scenario 'bottom up' transport, because $^{13}C$-depleted carbon passes through the upper ocean globally and is recorded in marine proxy records there, before entering the atmosphere (and being recorded in ice cores). The alternative scenario to explain the early deglacial decline in planktic (and shallow benthic) $\delta^{13}C$ we term 'top down'. This recognizes the importance of air-sea exchange in conveying an isotopic signal from the atmosphere to the ocean surface rapidly (on the order of 1 yr) and globally (e.g. Schmittner et al., 2013), followed by propagation of the $\delta^{13}C$ signal from surface to upper intermediate depths occurring on a multi-decadal to centennial timescale (Heimann and Maier-Reimer 1996; Broecker et al., 1985; Eide et al., 2017). Although these timescales allow for an atmospheric $\delta^{13}C$ decline to be propagated throughout the upper ocean, this 'top down' effect has been mostly overlooked in the interpretation of marine planktic and benthic $\delta^{13}C$ records, at least until recently (Lynch-Stieglitz et al., 2019).

The 'top down' scenario has very different implications from 'bottom up'. Firstly, negative $\delta^{13}C$ excursions recorded in the upper ocean need not be associated with enhanced influx of nutrients (based on the notion that the extra nutrients came from a previously isolated deep ocean reservoir

along with isotopically depleted respired metabolic carbon). Secondly, a 'top down' scenario does not require a specific or even a single initial path of carbon to the atmosphere. Outgassing to the atmosphere could occur anywhere at the ocean surface, with a negative $\delta^{13}C$ signal that then propagates globally through air-sea gas exchange – akin to the on-going fossil fuel $CO_2$ emissions and the propagation of its isotopically depleted signal down through the ocean (Eide et al., 2017).

In this paper we take a two-pronged approach to help elucidate the more likely of these end-member scenarios. Firstly, we present a new benthic $\delta^{13}C$ record from the western equatorial Pacific (WEP) at 566m depth that fills an important data gap from intermediate water depths in the Pacific basin. The site is located in the pathway of SAMW and AAIW to the upper tropical Pacific (Figure 1c) and is also shallow enough be sensitive to $\delta^{13}CO_2$ changes in the 'top down' scenario. Secondly, the early deglacial section of this record is interpreted with insights gained from analyzing a transient deglacial simulation conducted with the Earth system model LOVECLIM (Menviel et al., 2018). The specific LOVECLIM simulation we utilize starts with a scenario of excess respired carbon accumulated in a more stratified deep Ocean with reduced ventilation rates. Although it is not clear if such a glacial carbon scenario is correct (Cliff et al., 2021, Stott et al., 2021), we can still make use of the ability of the model to simulate how the ocean communicates stored carbon and its isotopic composition to the atmosphere during deglaciation (the focus of this paper).

In the transient LOVECLIM simulation, sequestered respired carbon from the deep and intermediate waters is ventilated through the Southern Ocean, leading to a sharp decline in $\delta^{13}CO_2$, consistent with ice core records. We evaluate the two different $\delta^{13}C$ transport scenarios by partitioning the simulated carbon pool and its stable isotope signature into a preformed ($DIC_{pref}$,

being the carbon that is transported passively by ocean circulation) and a respired (DIC$_{soft}$, the accumulated respired carbon since the water parcel was last in contact with the atmosphere) component. Because the LOVECLIM transient experiment does not explicitly simulate either preformed or respired carbon as additional numerical tracers, the respired carbon is instead estimated by apparent oxygen utilization (AOU) – the difference between oxygen saturation and simulated [O$_2$] (see section 2.4). If the 'top down' transport scenario was the mechanism responsible for the $\delta^{13}$C decline in marine proxy records from the upper 1000m depth, the preformed signal should dominate, while a regenerated signal would dominate in the 'bottom up' scenario. The carbon partitioning framework is not new - previous studies have used this framework to study the mechanisms that lead to lower glacial atmospheric CO$_2$ (Ito and Follows, 2005; Ödalen et al., 2018; Khatiwala et al., 2019) and processes that control $\delta^{13}$CO$_2$ and marine carbon isotope composition (Menviel et al., 2015; Schmittner et al., 2013). This diagnostic framework has also been applied to study the carbon cycle perturbation in response to a weaker Atlantic Meridional Overturning Circulation (AMOC) (Schmittner and Lund, 2015), albeit in experiments that were performed under constant pre-industrial conditions. However, new here is the application of a 2$^{nd}$ Earth System model (cGENIE (Cao et al., 2009)) to fully evaluate the AOU-based off-line approach against an explicit respired organic matter $\delta^{13}$C tracer.

## 2 Methods

After describing the new foraminiferal $\delta^{13}$C record in section 2.1, we summarize the LOVECLIM model and published deglacial transient simulation in section 2.2. We then summarize the cGENIE earth system modelling framework and deglacial experiments in section 2.3 before describing the $\delta^{13}$C tracer partitioning framework in section 2.4.

## 2.1 Stable Isotope Analyses and Age Model for Piston Core GeoB17402-2

The WEP piston core GeoB17402-2 (8°N, 126°34'E, 556m water depth) (Figure 1c) was recovered from the expedition SO-228. Planktic foraminiferal samples for [14]C age dating were picked from the greater than 250μm size fraction of sediment samples and were typically between 2 and 5mg. All new radiocarbon ages were measured at the University of California Irvine Accelerator laboratory. An age model (Figure S1) was developed for this core with BChron using the Marine20 calibration curve (Heaton et al., 2020) without any further reservoir age correction.

For benthic foraminiferal $\delta^{18}O$ and $\delta^{13}C$ measurements approximately 4-8 *Cibicidoides mundulus* (*C. mundulus*) were picked. These samples were cleaned by first cracking the tests open and then sonicating them in deionized water after which they were dried at low temperature. The isotope measurements were conducted at the University of Southern California on a GV Instruments Isoprime mass spectrometer equipped with an autocarb device. An in-house calcite standard (ultissima marble) was run in conjunction with foraminiferal samples to monitor analytical precision. The one standard deviation for standards measured during the study was less than 0.1‰ for both $\delta^{18}O$ and $\delta^{13}C$. The stable isotope data are reported in per mil with respect to Vienna Pee Dee Belemnite (VPDB).

## 2.2 LOVECLIM Deglacial Transient Simulation

The LOVECLIM model (Goosse et al., 2010) consists of a free-surface primitive equation ocean model (3° × 3°, 20 vertical levels), a dynamic–thermodynamic sea ice model, an atmospheric model based on quasi-geostrophic equations of motion (T21, three vertical levels), a land surface scheme, a dynamic global vegetation model (Brovkin et al., 1997) and a marine carbon cycle model

(Menviel et al., 2015). To study the sensitivity of the carbon cycle to different changes in oceanic circulation, a series of transient simulations of the early part of the last deglaciation (19-15ka) (Menviel et al., 2018) was performed by forcing LOVECLIM with changes in orbital parameters (Berger, 1978), changes in the freshwater surface balance as well as Northern Hemispheric ice-

sheet geometry and albedo (Abe-Ouchi et al., 2007), and starting from a LGM simulation that best fit oceanic carbon isotopic ($^{13}$C and $^{14}$C) records (Menviel et al., 2017).

The simulation we analyzed for this study is "LH1-SO-SHW" from Menviel et al, (2018). We briefly describe the applied forcing in this simulation: Firstly, a freshwater flux of 0.07 Sv was added to the North Atlantic between 17.6 ka and 16.2 ka, resulting in an AMOC shut down.

Secondly, a salt flux was added to the Southern Ocean between 17.2 ka and 16.0 ka to enhance Antarctic Bottom Water (AABW) formation. Due to its relatively coarse resolution, the model could mis-represent the high southern latitude atmospheric or oceanic response to a weaker North Atlantic Deep Water (NADW). Enhanced AABW could have occurred due to a strengthening of the SH westerlies, changes in buoyancy forcing at the surface of the Southern Ocean, opening of

polynyas, or sub-grid processes. Lastly, two stages of enhanced Southern Ocean westerlies are prescribed in the simulation at 17.2 ka and at 16.2 ka; this timing generally corresponds to Southern Ocean warming associated with two phases of NADW weakening during Heinrich Stadial 1 (Hodell et al., 2017). For more detail about this experiment, see Menviel et al., (2018).

We chose to focus our analysis on this particular simulation because 1) recent ice core records also

suggest enhanced SO westerly winds during Heinrich stadials (Buitzert et al., 2018); 2) "LH1-SO-SHW"  matches some of the important observations (e.g. ice core record of atmospheric $CO_2$ and $\delta^{13}CO_2$) better than the other scenarios presented in Menviel et al.,(2018); 3) the stronger SO wind

stress in "LH1-SO-SHW" leads to an increased transport of AAIW to lower latitudes, which could have impacted the intermediate depths of the global ocean.

## 2.3 cGENIE Simulations

The cGENIE Earth system model is based on a 3-D frictional geostrophic ocean circulation component, plus dynamic and thermodynamic sea ice components, and is configured here at a resolution of 36x36 horizontal grid with 16 vertical layers in the ocean. The configuration we employ here lacks a dynamical (GCM) atmosphere, with atmospheric transport fixed and provided via a 2-D energy-moisture balance model (Edwards and Marsh, 2005). The low-resolution ocean component and highly simplified atmospheric component make cGENIE much less computationally expensive to run than LOVECLIM. As well as facilitating multiple sensitivity experiments run to (deep ocean circulation) steady state to help partition and attribute carbon sources and pathways.

Ocean carbon storage analysis using the cGENIE model has previously utilized a range of preformed tracers, including those of phosphate ($P_{pref}$), dissolved inorganic carbon ($DIC_{pref}$), dissolved oxygen ($O_{2pref}$), and alkalinity (Ödalen et al., 2018). In the model, these are implemented by resetting the current value of the tracer at the ocean surface at each time-step, to the corresponding 'full' tracer, e.g. the value of $DIC_{pref}$ is set to that of surface ocean DIC. (Technically, an anomaly is applied to each preformed tracer at the ocean surface at each time-step, equal to the difference between the current bulk tracer value and the preformed tracer value (as opposed to simply directly setting the values equal in the code). Because in the numerical scheme, all fluxes, including those induced by ocean circulation and any preformed tracer anomalies, are calculated simultaneously and only summed and applied to update the tracer concentration field at the very

end of the model time-step, preformed tracer concentrations at the ocean surface and at the end of the time-step, never exactly equal those of the bulk tracer.) Thereafter, these tracers are carried conservatively by ocean circulation, with no loss or gain due to e.g. organic matter remineralization in the ocean interior.

We expand the diagnostic tracer capabilities of cGENIE here and additionally add $DIC_{soft}$, which

is the contribution to DIC form respired carbon. This is implemented as a tracer reset to zero at the ocean surface at each time-step, but which is incremented by an amount of DIC equal to the remineralization of both particulate and dissolved organic matter and including organic carbon 'reflected' (not preserved and buried) from the sediment surface. As for the preformed tracers, ocean circulation also acts on the distribution of $DIC_{soft}$ in the model. Figure S2 illustrates how

DIC is partitioned for the preindustrial steady state of Cao et al. (2009). Note that we do not explicitly simulate $DIC_{carb}$ (the contribution to DIC from dissolving $CaCO_3$, either in the water column or at the sediment surfce) as a $4^{th}$ tracer, but rather simply calculate it as the difference between DIC and $DIC_{pref} + DIC_{soft}$.

We also create a novel addition to the model – preformed and respired $^{13}C$ ($\delta^{13}C_{pref}$ and $\delta^{13}C_{soft}$,

respectively). These are implemented as $DIC_{pref}$ and $DIC_{soft}$, but for the concentrations of $DI^{13}C$. (In cGENIE, isotopes are carried explicitly as concentrations with delta ($\delta$) values only generated in conjunction with bulk concentrations for output (and more convenient input).) Figure S3 illustrates how the $\delta^{13}C$ signature of DIC is partitioned into explicitly simulated preformed and respired carbon components, and with $\delta^{13}C_{carb}$ (the contribution to $\delta^{13}C$ of DIC from dissolved

$CaCO_3$) again calculated by difference.

A full description of the cGENIE tracer scheme together with $\delta^{13}C$ tracer decomposition and attribution error analysis for both steady-state carbon cycling as well as under an idealized perturbation experiment, is available in the Supplement, with the pertinent insights summarized in Results.

Finally, we create a transient deglacial-like experiment using cGENIE, to approximately mimic some of the key features of a changing climate and carbon cycle simulated by LOVECLIM. Although AOU based errors in estimating the partitioning of respired vs. preformed $\delta^{13}C$ are already addressed via the idealized cGENIE steady-state and transient experiments (SI), decoupling in time of atmospheric $CO_2$ (and $\delta^{13}C$), surface climate, biological export, and the

large-scale circulation of the ocean (and especially the AMOC) across the deglacial transition, may induce a more complex evolution of AOU-based error. We address this by then calculating how the AOU-based error changes change in a deglacial-like cGENIE experiment. For this, we take a model configuration based on the idealized 'glacial' boundary conditions of Rae et al., (2020) (including increased zonal planetary albedo at high Northern Hemisphere latitudes and the orbital

configuration at 21 ka). Note, we did not attempt to achieve a glacial-like atmospheric $CO_2$ value for this spin-up, instead, we prescribed atmospheric $CO_2 = 278$ppm, $\delta^{13}CO_2 = -6.5‰$. The spin-up was run for 10,000 years. We then performed a deglacial transient simulation with time varying salt/freshwater flux into the North Atlantic and the Southern Ocean as well as wind stress forcing over the Southern Ocean (Figure S8). We ran this experiment with all the diagnostic tracers

described above.

**2.4 Separating $\delta^{13}C$ Anomalies into the Preformed ($\Delta\delta^{13}C_{pref}$) and Respired ($\Delta\delta^{13}C_{soft}$) Component**

The published (Menviel et al., 2018) transient LOVECLIM model experiment that we analyze here does not include the numerical tracers required to explicitly attribute the sources of any given change in $\delta^{13}$C in the model ocean. We hence make approximations from AOU calculated in the model experiment but assess the errors inherent in this by means of a set of experiments using a 2[nd] Earth system model – 'cGENIE' (Cao et al., 2009). This approach is detailed as follows (and expanded upon further in the Supplement).

We assume the following carbon isotopic mass balance:

$$\delta^{13}C * DIC = \delta^{13}C_{pref} * DIC_{pref} + \delta^{13}C_{soft} * DIC_{soft} + \delta^{13}C_{carb} * DIC_{carb} \quad (1)$$

where DIC, $DIC_{pref}$, $DIC_{soft}$, and $DIC_{carb}$, are the dissolved total inorganic carbon, the preformed, respired organic matter ('Csoft'), and dissolved (calcium) carbonate carbon pools, respectively. $\delta^{13}C_{pref}$, $\delta^{13}C_{soft}$, and $\delta^{13}C_{carb}$, are the corresponding isotopic signatures (as ‰) that contribute to the $\delta^{13}$C signature of DIC and it is changes in the $\delta^{13}$C of DIC that we assume foraminiferal records reflect.

Any given observed $\delta^{13}$C anomaly in the ocean can then be expressed as:

$$\Delta\delta^{13}C = \Delta(\delta^{13}C_{pref} * DIC_{pref} / DIC) + \Delta(\delta^{13}C_{soft} * DIC_{soft} / DIC) + \Delta(\delta^{13}C_{carb} * DIC_{carb} / DIC) \quad (2)$$

The terms on the RHS represent the contribution of the preformed, respired, and dissolved (carbonate) components to the overall $\delta^{13}$C change, respectively. Since the contribution of $CaCO_3$ dissolution is small in the upper 1000m (where GeoB17402-2 is located) in carbon cycle models (see also the Supplement), and since there is no $^{13}$C fractionation during $CaCO_3$ formation in the LOVECLIM model, the last term on the RHS can be neglected for the purpose of this study.

We use AOU to estimate respired carbon and its contribution to the $\delta^{13}C$ changes: $\Delta(\delta^{13}C_{soft} *$ $DIC_{soft} / DIC) = \Delta(\delta^{13}C_{soft} * AOU * R_{c:-o2} / DIC)$, where $\delta^{13}C_{soft}$ is estimated by the $\delta^{13}C$ of export

POC in the overlying water column, $R_{c:-o2} = 117:-170$.

This leads to:

$$\Delta\delta^{13}C = \Delta(\delta^{13}C_{pref} * DIC_{pref}/ DIC) + \Delta(\delta^{13}C_{soft} * AOU * R_{c:-o2} / DIC) \quad (3)$$

The anomaly, defined as the difference between 15 and 17.2 ka, can be expanded as:

$$\delta^{13}C^{15ka} - \delta^{13}C^{17.2ka} = \delta^{13}C_{pref}^{15ka} * DIC_{pref}^{15ka} / DIC^{15ka} - \delta^{13}C_{pref}^{17.2ka} * DIC_{pref}^{17.2ka} / DIC^{17.2ka} +$$

$$\delta^{13}C_{soft}^{15ka} * AOU^{15ka} * R_{c:-o2} / DIC^{15ka} - \delta^{13}C_{soft}^{17.2ka} * AOU^{17.2ka} * R_{c:-o2} / DIC^{17.2ka} \quad (4)$$

The AOU approach to estimate respired carbon content assumes that the oxygen content of surface waters always reaches equilibrium with the overlying atmosphere. However, studies have shown that this is not always the case, particularly for water masses formed in high latitudes (Bernardello et al., 2014; Ito et al., 2004; Khatiwala et al., 2019, Cliff et al., 2021). As a result,

AOU likely overestimates respired carbon content in the deep ocean. Additional errors associated with the AOU approach may result from the non-linear solubility of $O_2$ and respiration that does not involve $O_2$ consumption (i.e. through denitrification or sulphate reduction) (Shiller, 1981; Ito et al., 2004). However, to what extent these biases will affect the relative contribution of preformed and respired carbon pool on $\delta^{13}C$ anomaly in a carbon cycle

perturbation event has not to our knowledge previously been evaluated. To address this, we performed a deglacial transient simulation with cGENIE (see section 2.3) and then applied equation (4) to the output, with the results then compared with the values that are explicitly simulated by cGENIE. We also conducted a simplified (modern configuration based) analysis of

steady state and transient error terms (Figure S2-S7), which we include in full in the Supplement

and discuss briefly in the main text.

## 3 Results

The new GeoB17402-2 benthic $\delta^{13}$C record from the intermediate WEP documents a -0.3 to -0.4‰

decline during the early deglaciation (Figure 1d). Although the foraminiferal $\delta^{13}$C proxy can be

complicated by temperature and carbonate ion changes (Bemis et al., 2000, Schmittner et al., 2017),

and thus may not solely reflect seawater DIC $\delta^{13}$C changes, core-top patterns of benthic

foraminiferal $\delta^{13}$C are highly correlated with present-day seawater DIC $\delta^{13}$C (Schmittner et al.,

2017). The apparent lag between the onset of decline in benthic $\delta^{13}$C at site GeoB17402-2 (Figure

1d) and in $\delta^{13}CO_2$ appears to be due to the relatively large age model uncertainty below 154cm in

the GeoB17402-2 record (median age ~16.2yr), up to 1-2 kyr (2SD) (Figure S1). Despite this age

uncertainty, the new benthic record from the tropical Pacific captures a similar $\delta^{13}$C decline as

recorded from similar depth sites in the tropical/subtropical Atlantic and Indian Ocean (Lynch-

Stieglitz et al., 2014, 2019; Romahn et al., 2014).

To investigate whether the early deglacial $\delta^{13}$C decline observed at these sites in the upper ocean

is dominated by the preformed or respired component, we carried out an in-depth carbon cycle

analysis of the LOVECLIM transient simulation (Menviel et al., 2018). In response to the applied

freshwater input to the North Atlantic (Figure 2a), the AMOC significantly weakens from its

glacial state (Figure 2c). This has only a minor effect on the atmospheric $CO_2$ and $\delta^{13}CO_2$ (Figure

2d, 2e). In contrast, enhanced ventilation of AABW and AAIW driven by a combined freshwater

(Figure 2a) and wind-stress (Figure 2b) driven breakdown of stratification leads to an atmospheric

CO₂ increase of ~25 ppm and $\delta^{13}CO_2$ decline of -0.35‰ between 17.2 and 15 ka (Figure 2d, 2e). This is a consequence of stronger upwelling bringing $^{13}C$-depleted deep waters to the upper ocean with $\delta^{13}C$ generally decreasing by 0.2-0.3‰ at most locations in the upper 1000m (Figure 3a, 3d, 3g). In all sectors of the Southern Ocean below 400m depth, $\delta^{13}C$ increases by 0.1-0.2‰ due to stronger ventilation. Throughout the mid-depth North Atlantic, $\delta^{13}C$ decreases by more than 0.3-0.4‰ due to the AMOC weakening (Figure 3g). Finally, the LOVELCIM simulates a North Pacific deep water mass when AMOC slows down (Menviel et al., 2014) and this leads to stronger ventilation and +0.3-0.4‰ $\Delta\delta^{13}C$ in the North Pacific below 1000m depth (Figure 3a).

Decomposing the LOVECLIM $\Delta\delta^{13}C$ signal into the $\Delta\delta^{13}C_{soft}$ and $\Delta\delta^{13}C_{pref}$ component, we find that the entire water column of the Southern Ocean is characterized with a strong positive $\Delta\delta^{13}C_{soft}$ (indicting a loss of respired carbon) and a strong negative $\Delta\delta^{13}C_{pref}$ (Figure 3b, 3c, 3e, 3f, 3h, 3g). In the rest of the global upper ocean below 1000m depth, $\Delta\delta^{13}C_{soft}$ is negative but of a magnitude smaller than 0.1‰, whereas a 0.2-0.3‰ decrease in $\Delta\delta^{13}C_{pref}$ accounts for most of the $\Delta\delta^{13}C$ signal. In the deep Indo-Pacific, $\Delta\delta^{13}C_{soft}$ and $\Delta\delta^{13}C_{pref}$ show opposite signs, with the positive $\Delta\delta^{13}C_{soft}$ dominating the net $\Delta\delta^{13}C$ (Figure 3a-3f). In the deep North Atlantic, $\Delta\delta^{13}C_{soft}$ and $\Delta\delta^{13}C_{pref}$ are both negative (Figure 3h, 3i), leading to the largest decrease in $\Delta\delta^{13}C$ across the ocean basins (Figure 3g).

For comparison, Figure 4 shows the $\Delta\delta^{13}C$, $\Delta\delta^{13}C_{soft}$ and $\Delta\delta^{13}C_{pref}$ response in a similar deglacial-like transient simulation conducted with cGENIE (see section 2.3 and Figure S8) in which the respired and preformed components are explicitly simulated. The $\Delta\delta^{13}C$ patterns (Figure 4a, 4d, 4g) are qualitatively similar with that simulated by LOVECLIM (Figure 3a, 3d, 3g), albeit the magnitude of positive $\Delta\delta^{13}C$ in the deep Pacific and negative $\Delta\delta^{13}C$ in the deep North Atlantic are

larger in cGENIE (compare 3a with 4a and 3g with 4g). cGENIE does not simulate any large positive $\Delta\delta^{13}C_{soft}$ or negative $\Delta\delta^{13}C_{pref}$ in the Southern Ocean above 3000m depth (Figure 4), in contrast to the AOU-based results from LOVECLIM (Figure 3). In the North Atlantic, the magnitude of negative $\Delta\delta^{13}C_{soft}$ and $\Delta\delta^{13}C_{pref}$ are both larger in cGENIE compared to LOVECLIM.

To assess the potential errors associated with the AOU-based approach used to process the LOVECLIM output, we also calculated AOU-derived estimates of $\Delta\delta^{13}C_{soft}$ and $\Delta\delta^{13}C_{pref}$ for the cGENIE deglacial transient simulation (see section 2.4). The results suggest that throughout the mid-depth North Atlantic, the AOU-based $\Delta\delta^{13}C$ decomposition may introduce errors up to 0.3-0.4‰ under a weakening of the AMOC (Figure 5). In the Southern Ocean (south of 40°S), the AOU-based approach overestimates the magnitude of the positive $\Delta\delta^{13}C_{soft}$ and negative $\Delta\delta^{13}C_{pref}$ by 0.1-0.4‰ (Figure 5); the largest errors occur in the Pacific sector. Based on these results from cGENIE, we suggest the apparent $\Delta\delta^{13}C_{soft}$ and $\Delta\delta^{13}C_{pref}$ in the Southern Ocean shown in the LOVECLIM decomposition (Figure 3) are largely overestimated. Nonetheless, both cGENIE and LOVECLIM (after correcting the errors estimated from the cGENIE deglacial transient experiment, see Figure 5) show that the preformed component contributes -0.1 to -0.2‰ to the total $\Delta\delta^{13}C$ signal in the upper 1000m of the Southern Ocean. To the north of 40°S in the upper 1000m of the global upper ocean (except for the upper North Atlantic), the errors are relatively minor (generally much less than 0.1‰ in magnitude) and the AOU-based approach can provide a reasonably good estimate (Figure 5, also Figure S5, S7).

Finally, we further evaluate the errors inherent in the AOU-based approach to the decomposition of the different contributions to the $\delta^{13}C$ changes by means of a series of idealized steady-state and transient cGENIE experiments, described in the Supplement. From this we find that errors in

estimating $\delta^{13}C_{soft}$ arise both from errors in AOU (themselves composed of errors due to assuming air-sea equilibrium and because $O_2$ solubility increases nonlinearly with decreasing temperature) and from the assumption that the isotopic signature of carbon released by the remineralization of organic matter at any location in the ocean reflects that of carbon exported from the directly overlying ocean surface. The latter error turns out to be small in LOVECLIM as a consequence of

its relatively small (3‰) simulated latitudinal variability in organic matter $\delta^{13}C$, leaving the better understood AOU-driven error to dominate the net uncertainty in reconstructing $\delta^{13}C_{soft}$. As a further consequence of this, under idealized transient changes in climate and ocean circulation in cGENIE (see the Supplement), the AOU-induced error in $\delta^{13}C_{soft}$ is almost invariant throughout the uppermost ca. 500 m of the ocean, simply because the error in AOU itself is close to zero here.

This confirms the conclusions drawn from tracer comparisons made in deglacial cGENIE experiments that at the depth of GeoB17402-2, the AOU-based approach is relatively robust.

**4 Discussion**:

**4.1 Atmospheric $\delta^{13}C$ Bridge**

In the LOVECLIM model [13]C-depleted carbon is ultimately sourced from the respired carbon that

accumulated in the deep and intermediate waters during the glacial period as a consequence of the imposed weakened deep-water formation (Menviel et al., 2017). We show that in this scenario the isotopic signal is first transmitted to the atmosphere through strong outgassing in the Southern Ocean (Figure 6). The atmosphere then transmits the $\delta^{13}C$ signal to the rest of the global surface and subsurface ocean through air-sea gas exchange. An illustrative example is the simulated

transient $\delta^{13}C$ minimum event between 16.2 -15.8 ka in LOVECLIM (Figure 2c), which originates from the Southern Hemisphere and specifically from enhanced ventilation of AAIW (Figure 2a).

In the model, if the 'top down' scenario is true, the upper water masses away from the Southern Hemisphere would show similar magnitude of $\delta^{13}C_{DIC}$ changes as $\delta^{13}CO_2$. On the other hand, if the 'bottom up' scenario is true, a large negative $\delta^{13}C$ anomaly (of respired nature) should first appear in the South Pacific subtropical gyre (STGSP), as STGSP lies on the pathway between Southern Ocean water masses and those at lower latitudes. Then the signal would progressively spread to the tropics and finally reach the North Pacific. The negative $\delta^{13}C$ anomaly may also be gradually diluted along its pathway from the South Pacific to the North Pacific. However, in the LOVECLIM simulation, there is no $\delta^{13}C$ minimum in the upstream STGSP, while the atmosphere-like negative $\delta^{13}C$ anomaly appears in the EEP thermocline, the North Pacific subtropical gyre (STGNP) and North Pacific Intermediate Water (NPIW) simultaneously (Figure 7). In addition, the millennial-scale $\delta^{13}C$ evolution in these upper ocean water masses to the north of the equator exhibits a pattern of change that is similar to the atmosphere (Figure 7). The synchronized $\delta^{13}C$ changes therefore point to the dominant role of atmospheric communication rather than time-progressive oceanic transport of a low $\delta^{13}C$ signal in LOVECLIM.

In the LOVECLIM simulation, both millennial- and centennial-scale $\delta^{13}CO_2$ declines are the result of enhanced deep ocean and/or intermediate ocean ventilation originating in the Southern Ocean. Using the UVic Earth-System model, Schmittner and Lund (2015) showed that a slow-down of AMOC alone is able to weaken the global biological pump and lead to light carbon accumulation in the upper ocean and the atmosphere, without explicitly prescribing any forcing in the Southern Ocean. Despite the different prescribed forcing, $\Delta\delta^{13}C_{pref}$ also dominates the total $\Delta\delta^{13}C$ in the upper 1000m of the global ocean in the UVic experiment (See Figure 6 in Schmittner and Lund, 2015). Taken together, simulations by all three models suggest that any process that lowers atmospheric $\delta^{13}CO_2$ would have an influence on the global upper ocean $\delta^{13}C$. In fact, the same

phenomenon has been recurring since the beginning of the industrial era due to fossil fuel burning

- known as the Suess effect (Eide et al., 2017). The 'top down' scenario is also compatible with

the concept of a nutrient teleconnection existing between the Southern Ocean and low latitudes

(Palter et al., 2010; Pasquier and Holzer, 2016; Sarmiento et al., 2004). Figure 8 illustrates that

stronger upwelling brings excess nutrients to the surface of the Southern Ocean. Unused nutrients

are then transported to low latitudes within the upper ocean circulation (e.g. through mode waters

and thermocline waters). However, a nutrient teleconnection does not, in itself, reflect an enhanced

flux of $^{13}$C-depleted DIC from the deep ocean to low latitudes in a 'tunnel-like' fashion (and

'bottom up' transport).

In the following sections, we present two cases where the LOVECLIM transient simulation

successfully captures the early deglacial $\delta^{13}C_{DIC}$ evolution recorded in marine proxies. The model-

based $\Delta\delta^{13}C$ partitioning then offers a unique opportunity to investigate the controlling

mechanisms of the observed marine $\delta^{13}C$ variability. We acknowledge that there are also places

where models (in both LOVECLIM and cGENIE deglacial transient simulations) fail to simulate

the observed $\delta^{13}C$ trend between 17.2 and 15 ka. For instance, models simulate significant positive

$\Delta\delta^{13}C$ (above 0.4-0.5‰) (Figure 3a, 4a) in the deep tropical/North Pacific whereas observations

record no significant trend (Lund and Mix 1998, Stott et al., 2021). Models also simulate very

small $\Delta\delta^{13}C$ (~0.1‰) in the deep tropical/northern Indian Ocean (Figure 3d, 4d) whereas proxy

records document a distinct +0.3-0.4‰ trend (Waelbroeck et al., 2006, Sirocko et al., 2000). The

model-data disagreement in the deep Indo-Pacific warrants future study.

**4.2 Revisiting EEP Thermocline $\delta^{13}C$**

Waters at EEP thermocline depths are thought to be connected to the deep ocean through AAIW from the south and NPIW from the north. The EEP is therefore a potential conduit for deep ocean carbon release to the atmosphere. On the other hand, the EEP thermocline is also shallow enough to record an atmospheric $\delta^{13}C$ signal, either directly through gas exchange at the surface or indirectly through a preformed signal acquired from other parts of the global surface ocean. We select two EEP thermocline $\delta^{13}C$ records from different oceanographic settings (Figure 9a): site GGC17/JPC30 is near the coast, featured with relatively low surface nutrients; site ODP1238 is located in the main upwelling zone, with relatively high surface nutrients. Previous studies suggest the deglacial history of deep-water influence at the two sites were also distinctively different: at site ODP1238, strengthened deglacial $CO_2$ outgassing inferred from boron isotope data has been interpreted to reflect respired carbon transported from the Southern Ocean (Martínez-Botí et al., 2015); at site GGC17/JPC30, wood-constrained constant surface reservoir ages over the last 20 ka suggest this site was not influenced by old respired carbon from high latitudes (Zhao and Keigwin, 2018). However, the early deglacial planktic $\delta^{13}C$ records from the two sites show remarkably similar evolution, which is well captured by the LOVECLIM transient simulation (Figure 9b). By comparing Figure 3b to 3c, it is clear that the simulated $\delta^{13}C$ anomaly in the EEP thermocline (~100m) is dominated by the preformed component. The modeling evidence indicates that even though the EEP is the largest $CO_2$ outgassing region (in terms of absolute $\Delta pCO_2$, Figure S9) under an enhanced Southern Ocean upwelling scenario, its thermocline $\delta^{13}C$ is dominantly controlled by the 'top down' mechanism rather than the 'bottom up' mechanism as previously suggested (Martínez-Botí et al.,2015; Spero and Lea, 2002). The apparent conundrum can be explained by the fact that the air-sea balance of carbon isotopes is achieved through *gross* rather than *net* $CO_2$ exchange. Collectively, we make the case that in strong upwelling regions (e.g. the EEP) that are

remotely connected to the deep ocean, thermocline $\delta^{13}C$ is still subjected to strong atmospheric

overprint.

**4.3 How Deep in the Ocean Can the Negative $\Delta\delta^{13}C_{pref}$ Signal from the Atmosphere Penetrate During the Early Deglaciation?**

We have shown that given the dominant control of preformed $\delta^{13}C$ component in the upper ocean, some interpretations of planktic $\delta^{13}C$ records might need to be re-evaluated. Our

simulations also reveal that an atmospheric influence can reach deeper than thermocline depths and down to upper intermediate depths – consistent with what Lynch-Stieglitz et al., (2019) proposed. Below 1000m, a $\Delta\delta^{13}C_{pref}$ signal from the atmosphere may still exist, but no longer dominates the total $\Delta\delta^{13}C$ as $\Delta\delta^{13}C_{soft}$ becomes increasingly important at depth. (The contribution of $\delta^{13}C_{carb}$ also increases at depth (Figure S3) and can exceed 10% of the

contribution of $\delta^{13}C_{soft}$.)

It has been suggested that deglacial $\delta^{13}C$ variability in the waters above 2000m depth in the Atlantic could be driven by air-sea exchange (Lynch-Stieglitz et al., 2019). However, mid-depth (1800-2100m) benthic $\delta^{13}C$ records from the Brazil margin (~27°S) document a sharp decline of 0.4‰ at ~18 ka (Lund et al., 2019), while atmospheric $\delta^{13}CO_2$ did not decrease until ~17 ka

(Bauska et al., 2016; Schmitt et al., 2012). Lund et al., (2019) argued that the lagging atmospheric $\delta^{13}CO_2$ decline seemed at odds with the idea that $\delta^{13}C_{pref}$ contributed to the early benthic $\delta^{13}C$ decrease at their site. The observed benthic $\delta^{13}C$ trend between 20-15 ka at these Brazil margin sites is well simulated by LOVECLIM (Figure 10), allowing us to explore this question further. Before atmospheric $\delta^{13}CO_2$ starts to decline in LOVECLIM at ~17.2 ka, changes in $\delta^{13}C_{DIC}$ at

~2000m depth at the Brazil Margin are dominantly controlled by excess accumulation of respired

carbon (indicated by highly negative $\Delta\delta^{13}C_{soft}$, Figure S10b), itself a response to the weakened AMOC, while $\Delta\delta^{13}C_{pref}$ is relatively small (Figure S10c). This is consistent with what previous studies have suggested (Lacerra et al., 2017; Lund et al., 2019; Schmittner and Lund, 2015). Interestingly, LOVECLIM also reveals a strong negative $\Delta\delta^{13}C_{pref}$ signal between 17.2 and 15 ka when atmospheric $\delta^{13}CO_2$ declines (Figure 3i). However, a positive $\Delta\delta^{13}C_{soft}$ (Figure 3h) signal originating from a loss of respired carbon due to enhanced ventilation at those depths almost completely compensates for the negative $\Delta\delta^{13}C_{pref}$, which leads to virtually no net change in $\delta^{13}C_{DIC}$ in the simulation (Figure 3g), consistent with the proxy observations (Figure 10). These results suggest that, between 17.2 and 15 ka, a negative preformed $\delta^{13}C$ signal from the atmosphere needs to be considered when interpreting benthic $\delta^{13}C$ records from the upper 2000m of the South Atlantic. The complexity associated with interpreting marine $\delta^{13}C$ records further underscores the urgent need to develop more robust means of estimating respired carbon accumulation/release from water masses.

**5 Conclusions:**

A transient simulation conducted by the LOVECLIM Earth system model is used as a realization of plausible pathways of low $\delta^{13}C$ signal transport under a prevailing deglacial scenario that involves Southern Ocean processes. Applying an AOU-based partitioning of carbon isotopic changes into preformed and respired components – a methodology that we scrutinize via a series of additional cGENIE Earth system model experiments – we show that ocean-atmosphere gas exchange likely dominates the negative $\delta^{13}C$ anomalies documented in global planktic and intermediate benthic $\delta^{13}C$ records between 17.2 and 15 ka. Numerical simulations further suggest that enhanced Southern Ocean upwelling can transfer $\delta^{13}C$ signals from respired carbon in the deep

ocean directly to the atmosphere. Consequently, atmospheric $\delta^{13}CO_2$ declines and this leaves its

imprint on the rest of the global upper ocean through air-sea exchange. The preformed component

dominates the upper 1000m and could account for a 0.3-0.4‰ decline in marine $\delta^{13}C$ records

during the early deglaciation, whereas the respired component becomes increasingly important at

deeper depth. At the same time, the amount of upwelling in the Southern Ocean is a forcing

imposed on the model rather than directly constrained. It is therefore possible there were other

sites where excess carbon was ventilated to the atmosphere during the deglaciation, which would

have also affected $\delta^{13}CO_2$. Our findings imply that planktic and upper intermediate benthic $\delta^{13}C$

records do not provide strong constraints on the site or the mechanisms through which $CO_2$ was

released from the ocean to the atmosphere. Interpretations of early deglacial upper intermediate

depth benthic $\delta^{13}C$ records also need to take into account an atmospheric influence. Whereas in

the model simulations the source of the atmospheric signal is a direct response to enhanced

Southern Ocean upwelling, our results underscore the need to find a way to fingerprint the actual

source(s) of $^{13}C$-depleted carbon that caused the atmospheric $\delta^{13}CO_2$ decline.

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

**Competing interests.** The authors declare that they have no conflict of interest.

**Acknowledgements.** This work is supported by the National Science Foundation under Grant Numbers 1558990 (to J.S. and L.S.) and 1736771 (A.R.). A.R. additionally recognizes support from the Heising Simons Foundation. L.M. acknowledges funding from the Australian Research Council grant FT180100606. The funding for the expedition SO-228 is from BMBF (German Ministry of Education and Research) grant #03G0228A. We thank Fortunat Joos and another anonymous reviewer for their constructive comments and prompting that helped to significantly improve this manuscript.

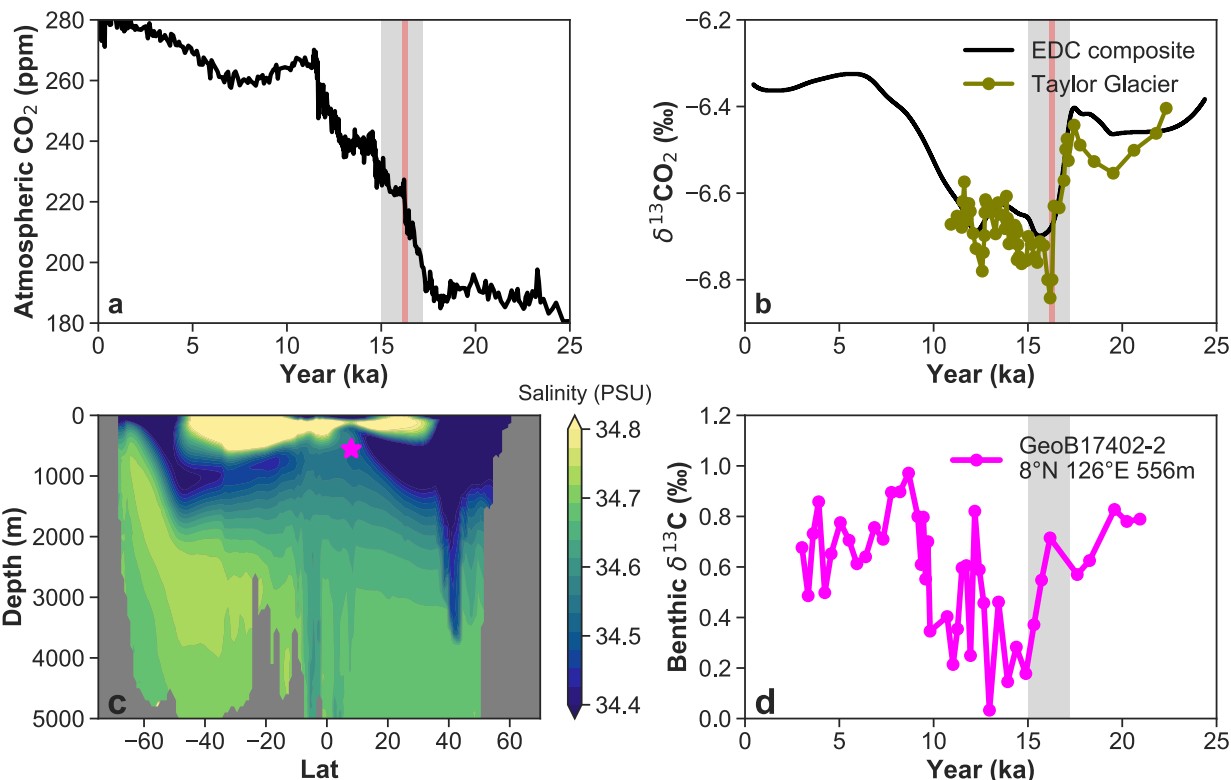

**Figure 1. a) Ice core records of atmospheric CO₂ (Bereiter et al., 2015; Marcott et al., 2014).**
**b) δ¹³CO₂ records (Bauska et al., 2016; Schmitt et al., 2012). c) WOA-18 Pacific zonal mean**
**(120-160°E) salinity, the magenta star marks the GeoB17402-2 site.  d) *C. mundulus* δ¹³C**
**record for upper intermediate depth and mode waters in the western equatorial Pacific. The**
**millennial- and centennial-scale events in these records are highlighted in grey and red,**
**respectively.**

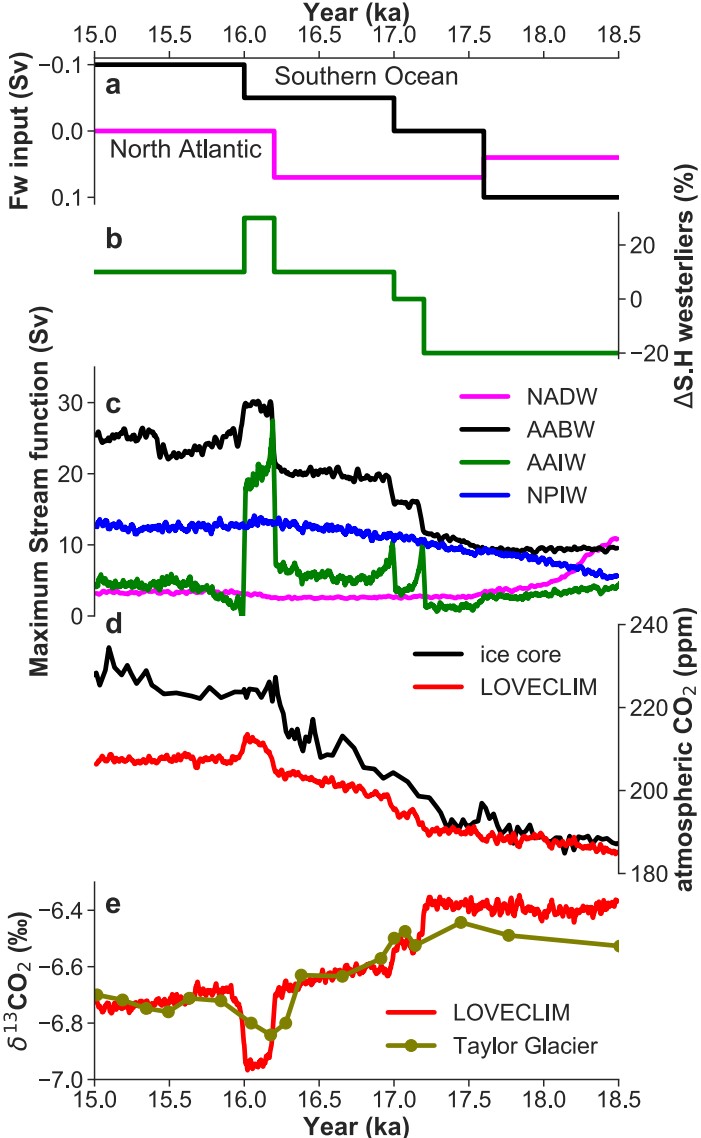

**Figure 2. Timeseries from the LOVECLIM transient experiment (Menviel et al., 2018). a) Freshwater input into the North Atlantic and the Southern Ocean; b) Southern Hemisphere westerly wind forcing; c) simulated NADW, AABW, AAIW and NPIW maximum stream function in LOVECLIM. 21-year moving averages are shown for the maximum stream function to filter the high-frequency variability; d) Ice core record of atmospheric $CO_2$ (Bereiter et al., 2015; Marcott et al., 2014) and LOVECLIM simulated atmospheric $CO_2$; e) The Taylor glacier $\delta^{13}CO_2$ record (Bauska et al., 2016) and LOVECLIM simulated $\delta^{13}CO_2$.**

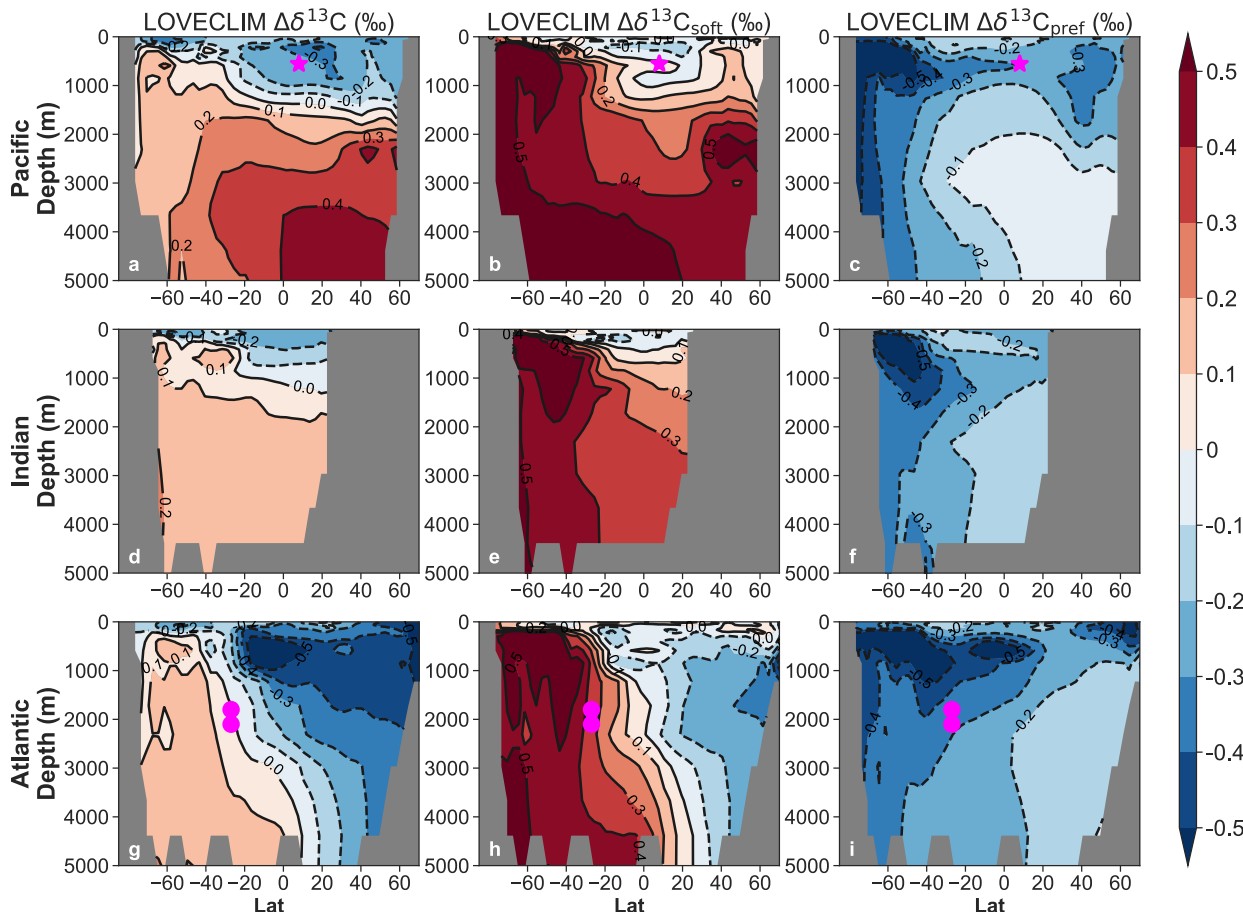

**Figure 3. Ocean basin zonal mean anomalies (15ka minus 17.2ka) as simulated in LOVECLIM. Top row: Pacific zonal mean anomaly (160°E-140°W). The magenta star marks the GeoB17402-2 site. Mid row: Indian zonal mean anomaly (50-90°E). Bottom row: Atlantic zonal mean anomaly (60°W-10°W). The magenta circles mark the 78GGC and the 33GGC site discussed in section 4.3.**

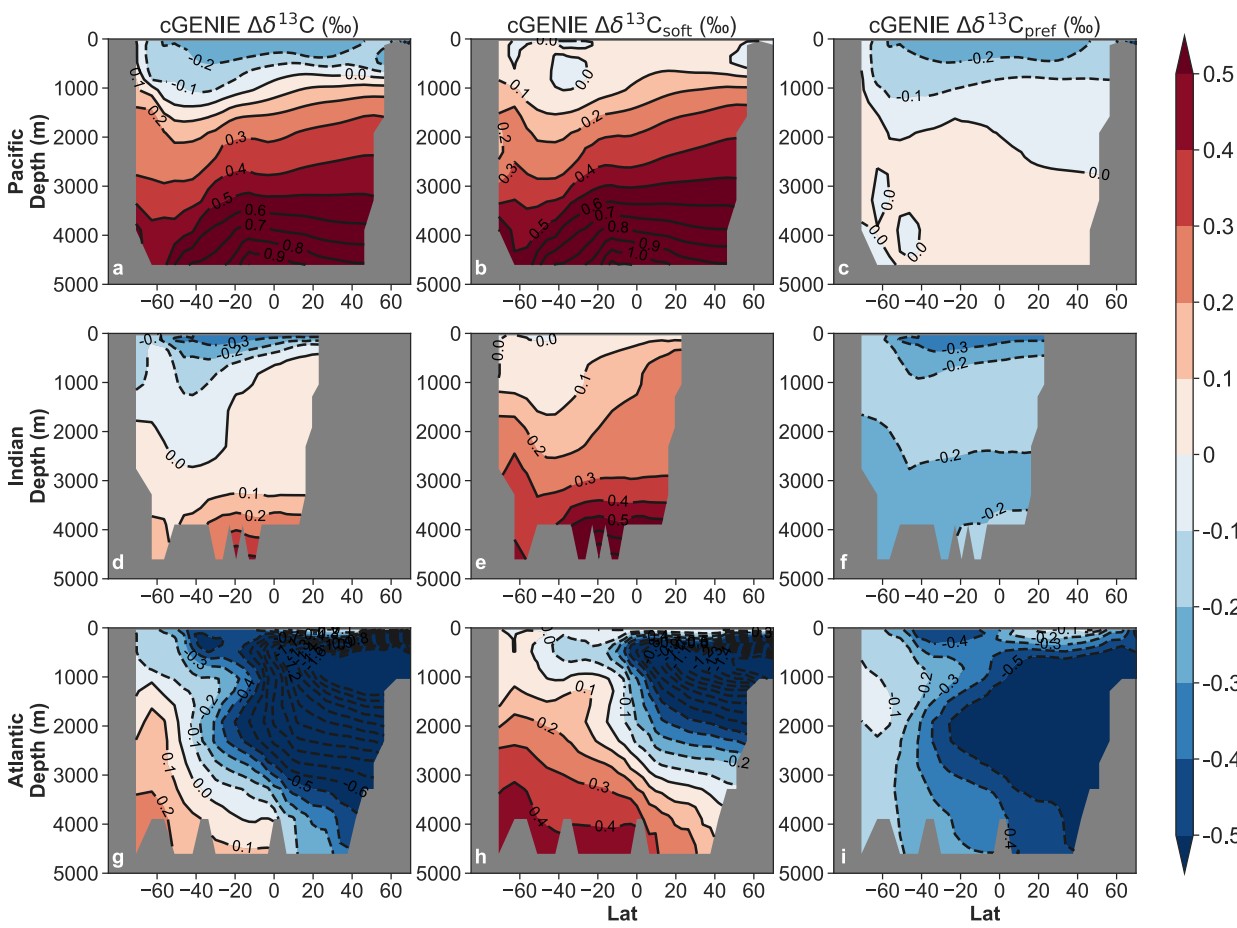

**Figure 4. Ocean basin zonal mean anomalies (15ka minus 17.2ka), but for the cGENIE deglacial transient simulation. Panels are organized as in Figure 3.**

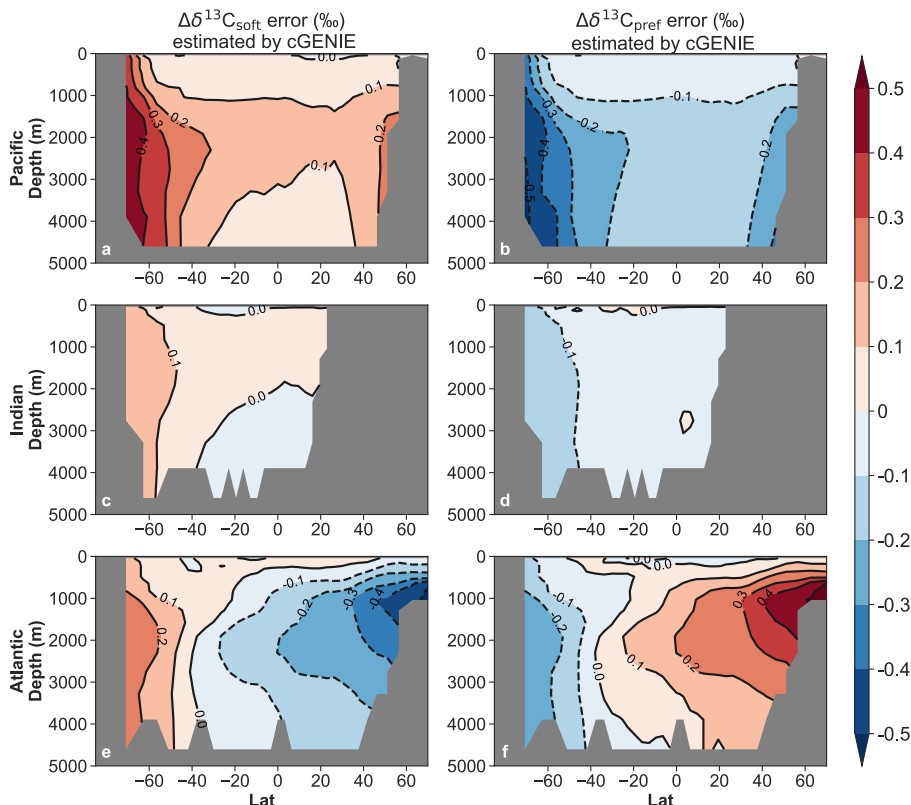

**Figure 5. cGENIE early deglacial transient AOU error analysis for Δδ¹³C_soft (left column)
and Δδ¹³C_pref (right column). The anomalies are defined as 15ka minus 17.2ka. The errors
are defined as AOU-based anomaly minus explicitly simulated anomaly.**

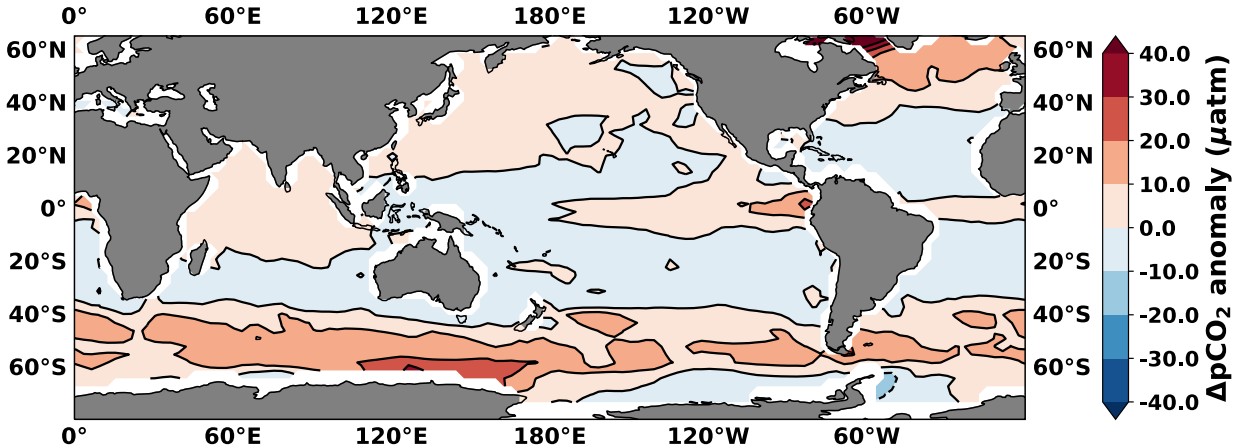

**Figure 6. Changes in air-sea pCO₂ gradient (15ka minus 17.2ka) simulated by LOVECLIM.**

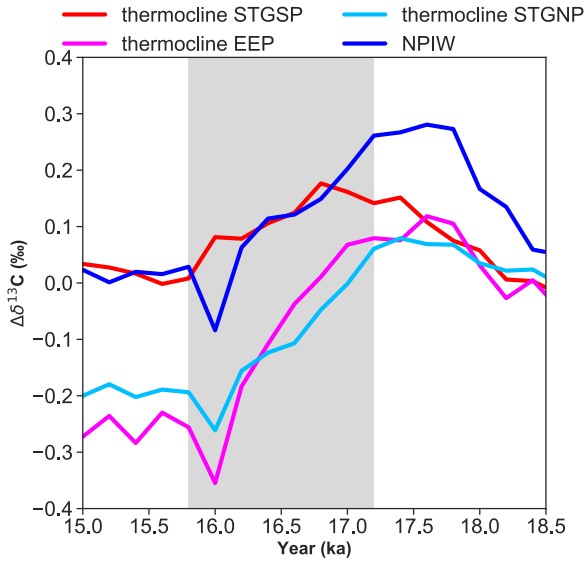

780

**Figure 7. LOVECLIM simulated $\Delta\delta^{13}C$ in thermocline EEP (90-82°W, 5°S-5°N, 77-105m), South Pacific subtropical gyre (STGSP, 160°E- 100°W, 40-22°S, 187-400m), North Pacific subtropical gyre (STGNP, 110°E- 140°W, 22-40°N, 187-400m), NPIW (167-170°E, 54-57°N, 660m. The average of 23.8-20 ka (i.e. LGM) is used as a reference level for the $\Delta\delta^{13}C$ calculations. The interval of decreasing $\delta^{13}C$ is highlighted with a grey bar.**

785

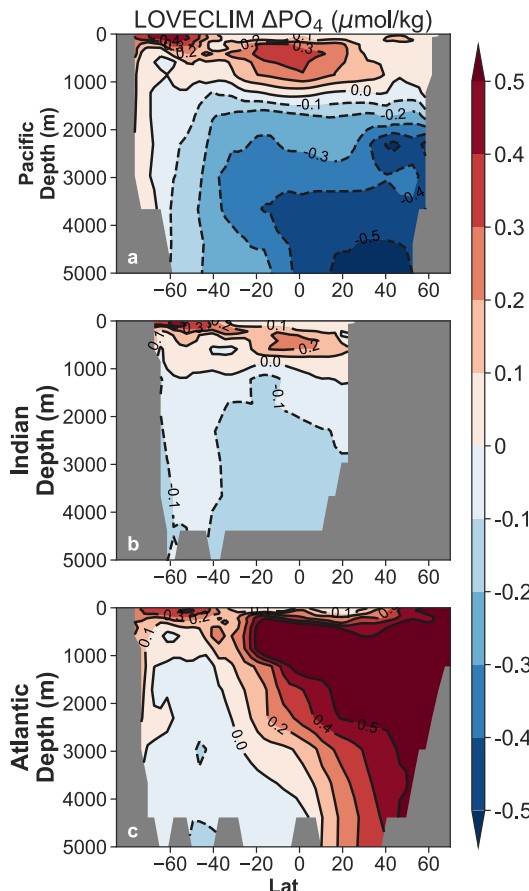

**Figure 8. Ocean basin zonal mean PO₄ anomalies (15ka minus 17.2ka) as simulated in LOVECLIM.**

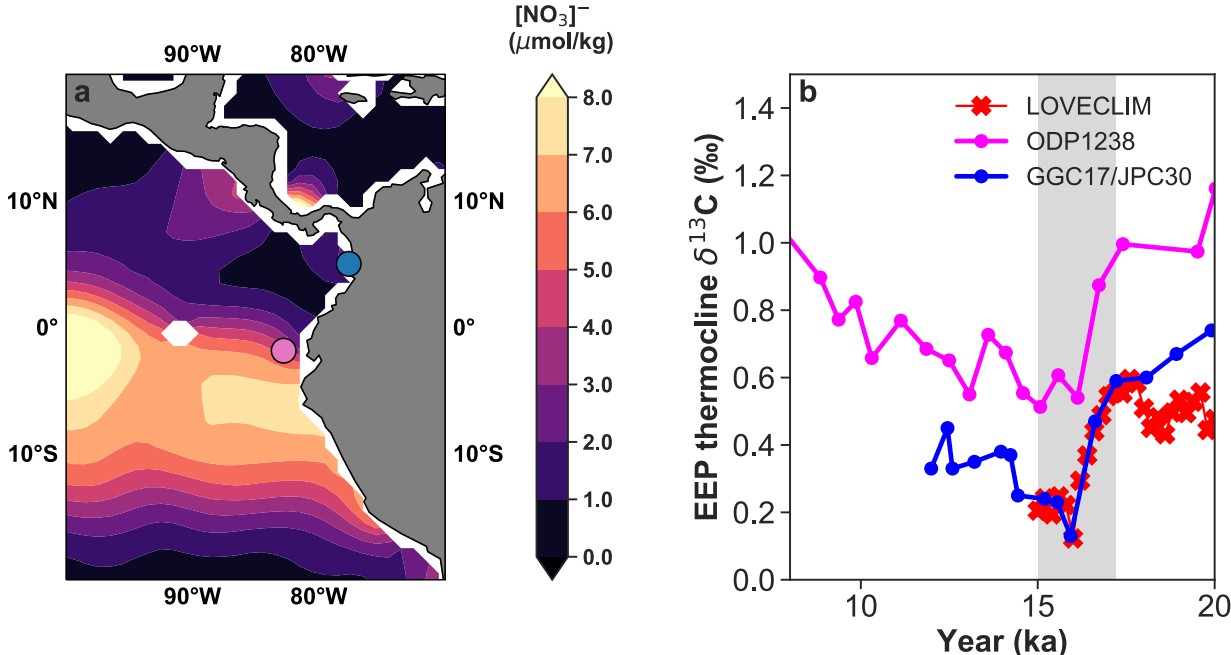

**Figure 9. a): Modern sea surface nitrate concentration from the WOA 18 dataset. The site of ODP 1238 and GGC17/JPC30 are marked as a purple and blue circle, respectively. b): *Neogloboquadrina. dutertrei* (*N. dutertrei*, a shallow thermocline species) δ¹³C data from ODP 1238 (Martínez-Botí et al., 2015), GGC17/JPC30 (Zhao and Keigwin, 2018), and LOVECLIM simulated δ¹³C of DIC at 100m (average of 82-90°W, 5°S-5°N). The *N. dutertrei* data are corrected by -0.5‰ to normalize to δ¹³C of DIC (Spero et al., 2003). The grey shaded bars highlight the time period we focus in this study.**

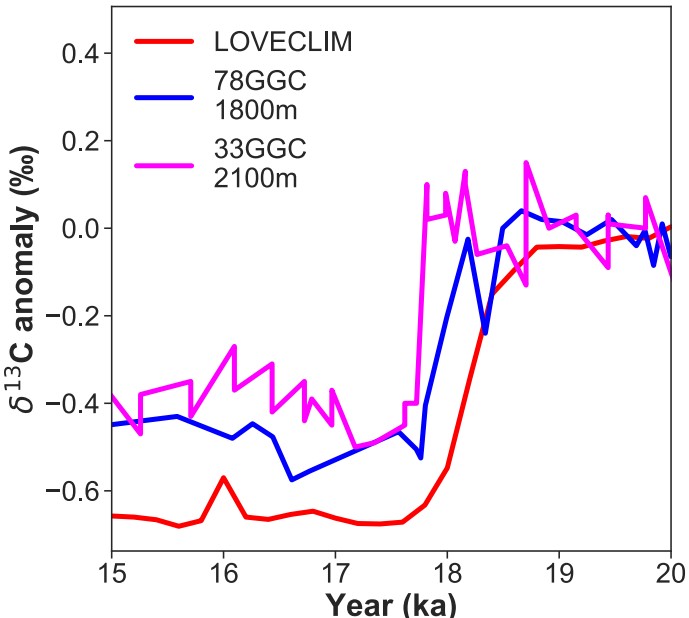

**Figure 10. Observed δ¹³C anomaly of 78GGC and 33GGC from the mid-depth of Brazil**

800     **Margin at 27°S (Lund et al., 2015) and LOVECLIM simulated δ¹³C anomaly at this location.**