# Peer review of "The Atmospheric Bridge Communicated the $\delta^{13}$ C Decline during the Last Deglaciation to the Global Upper Ocean"

_Climate of the Past, 2020_

## Referee Comment (RC1) · Anonymous Referee #1 · 14 Sep 2020

Shao et al. assess the mechanisms driving changes in the stable carbon isotopic composition of the upper ocean and in atmospheric CO2 (d13CO2) during the last deglaciation, focusing on the first major decline in d13CO2 observed in Antarctic ice core records around 17 kyr before present. Based on model simulations with LOVE-CLIM and GENIE, the authors test two hypotheses that may explain these trends: first, the upwelling of respired carbon (with a low-d13C signature) from the deep ocean, primarily in the Southern Ocean and its advection to the global ocean via the thermocline, and subsequent equilibration with the atmosphere (bottom-up scenario); and second, the sub-surface supply of respired carbon and strong equilibration with the atmosphere in upwelling regions (causing a decrease in d13CO2), and parallel transmission of the

atmospheric d13CO2 signal to the upper ocean via air-sea gas exchange (top-down scenario). Through a carbon speciation analysis, the authors find a strong influence of the top-down process on global upper-ocean d13C records (including a new one from the western equatorial Pacific), confirming important proxy-based postulations made by Lynch-Stieglitz et al. (2019).

This paper is a timely model-study on the mechanisms of global d13C records, testing (opposing) inferences on the global carbon cycle made initially by Spero and Lea (2002) and more recently by Lynch-Stieglitz et al. (2019). It therefore merits publication in Climate of the Past. I do, however, have difficulties to follow the argumentation of the authors in places, see why different model approaches were chosen (transient vs. equilibrium, glacial vs. interglacial boundary conditions) and whether they are appropriate for the premise of the study (in particular, their combination). The study essentially confirms the proposition of Lynch-Stieglitz et al. (2019) but I see some scope to provide novel insights that would increase the impact of the study. I elaborate on these aspects and other minor ones below. I recommend major revisions of the paper prior to publication. I also want to sincerely apologize to the authors for the delay in providing my evaluation of their manuscript. I hope that despite the delay the authors find my comments useful in improving their study.

Major comments:

Preformed and remineralized speciation in Introduction: The partitioning of ocean carbon into 'preformed' and 'remineralized' is central to the authors' study, but these important terms are not properly introduced in the study. A definition of these terms in the introduction are needed, and in particular how they are defined and what processes they are influenced by in the real world and in the model world. The latter I find somewhat incomplete: How do kinetic equilibration effects play into the partitioning process of carbon between the atmosphere and ocean, besides thermodynamic equilibration effects and primary production? Are surface wind effects considered as drivers of air-sea gas exchange in the model? Through the impact of surface wind stress on the

piston velocity or gas transfer coefficient, winds have a strong influence of air-sea gas exchange in the real world (e.g., Wanninkhof, 1992). Also note in line 59, that changes in the residence time of water parcels at the surface can also lead to preformed carbon changes, simply by varying the time available for air-sea gas exchange. This statement needs to be revised accordingly. Line 83. Justification is needed why the simulation LH1-SO-SHW was chosen although Menviel et al. (2018) provide a number of other simulations with increase Southern Ocean ventilation, e.g. LH1-SO.

Offline calculations of carbon species in LOVECLIM: I find it striking that the authors' "approach requires accurate representation of the preformed and remineralized components" (line 62), but that the LOVECLIM model does not simulate them explicitly. The authors need to discuss what types of errors might affect their offline calculation based on the LOVECLIM and how large these errors might be. For instance, why does AOU overestimate true oxygen utilization? (line 182). I find the sensitivity experiments made in cGENIE to alleviate the problems associated with the necessity of an offline calculation not convincing, because the experimental setup, forcing and boundary conditions are very different. This leads to my next point of criticism.

Comparability and suitability of LOVECLIM and cGENIE simulations: How do the cGENIE and LOVECLIM simulations support each other, when they are so different? Is it correct that wind changes are not considered in the cGENIE simulation (which they are in the LOVECLIM simulation)? If correct, this should be clearly stated. In that case, would this call for the use of LH1-SO instead of LH1-SO-SHW? How preformed nutrients or carbon are simulated in cGENIE is unclear, in particular given the statement in line 98 to 99. If preformed tracer values are reset to the full tracer value (what is this?) at each model step, does this skew the outcome to a dominance of preformed changes? I believe some more explanation is required here, as this suggests that all water masses leaving the surface ocean, e.g. in the Southern Ocean, have no remineralized tracer component.

Relative contributions of top-down and bottom-up processes: The authors suggest

that air-sea gas equilibration leaves a strong imprint on upper-ocean d13C records, while also acknowledging that bottom-up processes cannot be neglected, more so in some regions over others (e.g., line 174-179). However, the authors focus a lot on the top-down process, while in my view they would be in the position of *quantifying* what the relative contributions of these different processes in *different regions* are (and provide a global map accordingly). This would significantly increase the impact and value of the study, in particular for those researchers working with proxy data. I hence encourage the authors to consider performing these analyses. The study should also better highlight the finding that upper-ocean d13C are ultimately affected by both (top-down and bottom-up) processes but with strongly varying proportions in different regions.

Focus on initial deglacial d13CO2 decline: It is confusing that in places the entirety of the deglacial d13CO2 is discussed although boundary conditions and driving mechanisms might differ throughout the deglaciation (e.g. 162-164). I recommend to remove these and instead exclusively focus on the early deglacial d13CO2 change. The same (somewhat) applies to the centennial change in pCO2 around 16.2 kyr before present (e.g., 206-208).

Representation of foraminiferal d13C of true DIC d13C changes: It might be worthwhile to highlight in the manuscript that the one-to-one representation of seawater DIC d13C changes based on foraminiferal d13C is imperfect, more so for planktonics than for benthics (e.g., Bemis et al., 2000; Schmittner et al., 2017). It might be hence useful to clarify whether the trends and/or the magnitude of benthic d13C change resembles atmospheric d13C changes, e.g., in line 261, and whether both can be linked without reservations.

Minor comments:

Line 23. Specify the depths that relate to "from depths that are potentially affected by the atmosphere".

Line 28. I find that the statement "The mechanisms and the chain of events that were responsible for this pCO2 are not well understood" neglects a large body of literature, a large number of existing hypotheses and a wealth of proxy-data in support of some of these. I recommend some more nuance and adjustments to reflect this. E.g. "Despite xxx, the mechanisms ..." or "Although the leading hypothesis for millennial- and centennial-scale pCO2 rise was suggested to be xxx, the chain of events ..."

Line 44. I do not think that a clear lead of a d13CO2 decline can be or was documented. I hence recommend removing "initially occurring in the atmosphere"

Line 51. The statement "and the subsequent d13C decline ..." needs to be revised as it is confusing. How can a d13C decline contribute to pCO2 variability? I recommend changing it to "is a reflection of the evasion of oceanic carbon to the atmosphere, contributing to ..."

Line 63. Specify what components.

Line 70. "To our knowledge, the origin .." this sentence is confusing and seems out of place. Please revise.

Line 72. It is entirely unclear at this stage why a new benthic d13C record has been obtained. This sentence should be moved or the premise of these analyses should be introduced.

Line 87. Insufficiencies of the models in representing sub-grid processes are unquestionable. This statement should not be phrased as if they were not.

Line 108. It is entirely unclear why the forcing is limited to the Pacific sector of the Southern Ocean. Please specify. Here for consistency, I recommend changes a similar forcing to Menviel et al., (2018).

Line 120. A full sentence is needed here. Also, DICorg is depleted in 13C not d13C.

Line 121. Budget of what?

Line 123: (Dd13Creg) instead of (Dd13C)

Line 124: Is d13Corg defined or simulated? Is DIC =DIC total, i.e. reg + preformed? How is 12Corg defined?

Line 129. 2 and 5 mg CaCO3.

Line 131-132. What suggests that there is no evidence for invariable surface ocean reservoir age changes over the deglaciation? It is not enough to say that. I believe it has to be justified. Also Figure 4 shows a marked lag between the onset of d13C decline in the GeoB17402 and in atmospheric d13CO2. Is this real or an artifact of the age model (i.e., variable reservoir ages?)? I am surprised that there is no mention/discussion of this lag in the study.

Line 133. Remove "Once the calendar ages were established the results were plotted vs depth."

Line 140. Remove "will be archived in Pangaea" and add URL to appropriate section Data availability.

Line 142-143. Remove "Below.. " I don't find this helpful here, and the structure of the manuscript can be reflected in the headings.

Line 149. Which model?

Line 152-154. I am surprise to see a discussion of entirely new carbon species/terms (Dd13Cthermo and Dd13Cres), which haven't been introduced or mentioned earlier. They need to be properly introduced, otherwise this analysis is entirely confusing, and not very helpful for the reader. They also appear not to be of relevance throughout the remainder of the manuscript, which somewhat questions whether this analysis is needed. It is difficult to follow the statements in the following lines 154- 157: What is meant here with Dd13C? What does the preformed signal reflect? Dd13Cthermo? Please clarify.

[Figure]

Line 165. It should be pointed out clearly what observations lead to this major finding.

Line 172. "evolution" instead of "pathway"

Line 188. The d13C decline in the upper 1000 m (where? Does Figure 6 show a global ocean mean?) is also dominated by the preformed signal (everywhere?). Also some more help and explanation with regards to Figure 6 is needed, as it shows four panels.

Line 215-217: Reference to a figure is required.

Line 277-280: Please specify what time interval you refer to here. This also seems like an add-on that is not properly analyzed, and I hence wonder how useful this is. The authors would be in the position to test the different hypotheses of why the Atlantic and Pacific anomalies are so different, but that is entirely glossed over at this stage.

References

Bemis, B.E., Spero, H.J., Lea, D.W., Bijma, J., 2000. Temperature influence on the carbon isotopic composition of Globigerina bulloides and Orbulina universa (planktonic foraminifera). Mar. Micropaleontol. 38 (3), 213–228. https://doi.org/10.1016/S0377-8398(00)00006-2

Lynch-stieglitz, J., Valley, S.G., Schmidt, M.W., 2019. Temperature-dependent ocean–atmosphere equilibration of carbon isotopes in surface and intermediate waters over the deglaciation. Earth Planet. Sci. Lett. 506, 466–475. https://doi.org/S0012821X18306836

Menviel, L., Spence, P., Yu, J., Chamberlain, M.A., Matear, R.J., Meissner, K.J., England, M.H., 2018. Southern Hemisphere westerlies as a driver of the early deglacial atmospheric CO2 rise. Nat. Commun. 9 (1), 2503. https://doi.org/10.1038/s41467-018-04876-4

Schmittner, A., Bostock, H.C., Cartapanis, O., Curry, W.B., Filipsson, H.L., Galbraith, E.D., Gottschalk, J., Hoogakker, B., Jaccard, S.L., Lisiecki, L.E., Lund, D.C., Martínez-

Méndez, G., Lynch-Stieglitz, J., Mackensen, A., Michel, E., Mix, A.C., Oppo, D.W., Peterson, C.D., Repschläger, J., Sikes, E.L., Spero, H.J., Waelbroeck, C., 2017. Calibration of the carbon isotope composition ($\delta$13C) of benthic foraminifera. Paleoceanography 32, 512–530. https://doi.org/10.1002/2016PA003072

Spero, H.J., Lea, D.W., 2002. The cause of carbon isotope minimum events on glacial terminations. Science 296 (5567), 522. https://doi.org/10.1126/science.1069401

Wanninkhof, R.H., 1992. Relationship between wind speed and gas exchange. J. Geophys. Res. 97 (92), 7373–7382. https://doi.org/10.1029/92JC00188

---

## Referee Comment (RC2) · Fortunat Joos (Referee) · 2 Oct 2020

Fortunat Joos

The paper forcefully demonstrates that air-sea gas exchange effectively acts to couple atmospheric and upper ocean d13C. This point is, - from a modelling point of view and for all those monitoring the penetration of the anthropogenic perturbation into the ocean - rather trivial and not new. The timescale to bring the surface layer in equilibrium with a d13C perturbation in the atmosphere by air-sea gas exchange is of order 10 years as outlined by Broecker, Peng and others. Numerous measurements of CFCs, bomb-produced radiocarbon, DIC, and notably of d13C demonstrate that the atmospheric perturbation in these tracers is communicated by air-sea gas exchange to the surface layer and by surface-to-deep exchange to deeper layers within years to decades (e.g., (Heimann and Maier-Reimer 1996;Broecker et al. 1985;Eide et al. 2017). Thus it is clear from an observational as well as from a modelling point that air-sea gas exchange is important and needs to be considered when addressing carbon isotopes. Unfortunately, the role of air-sea exchange is sometimes neglected in the interpretation of marine planktonic d13C records. It may therefore be appropriate to recall this point for the paleoceanographic tracer community.

Interesting is that the authors offer a quantification of the influence of preformed versus remineralized changes in d13C. However, the method applied to separate changes in d13C into the contribution from preformed sources and sources from biogenic particles is unclear and may not be appropriate (see below). Another interesting point, which deserves a bit more discussion, is the information on the change in d13C versus the change in atmospheric CO2 in response to an increase in deep ocean ventilation (AABW, AAIW) forced by prescribed changes in salt and Southern Ocean wind stress.

I recommend major revisions.

Specific comments:

1) Attribution of the d13C changes.

Section 2.3 The separation into preformed and remineralized d13C from the model output of DIC and d13C appears problematic. The assumption and simplifications of the approach are not explained to the readers. It would be preferable to simulate the preformed tracers online.

a) The authors use the equations given in line 124 to estimate the change in $\delta$13C due to a change in remineralization. However, the equation is unclear. A new term is used in this equation: "12Corg". I guess 12Corg should read DICorg, the amount of remineralized carbon. Then the equation given by the authors reads:
$\delta$13Cr=d13Corg * $\Delta$(DICorg /DIC)                    (1)

b) How is $\delta^{13}$Corg computed? I get the impression that a constant $\delta^{13}$C signature of organic carbon ($\delta^{13}$Corg) is assumed in the approach applied to distinguish the preformed and remineralized components in LOVECLIM.

c) Any perturbation in surface $\delta^{13}$C is also transferred to newly formed organic matter and CaCO3 and finally to remineralized carbon. In addition, changes in surface CO2 affect the fractionation factor for organic matter formation and thereby again d13Corg. Further changes in surface d13C also affect changes in the signature of preformed fluxes. It appears that changes in the isotopic signature of preformed and remineralized carbon fluxes are neglected. This seems an oversimplification.

d) The equation on line 124 used to compute the change in remineralized $\delta$13C needs to be properly derived. The mass balance should be considered in the separation of the different components. I distinguish preformed DIC (DICp; index p) and remineralized DIC (DICr; index r) and related fluxes (Fp, Fr).
Changes in $\delta$13C can arise due to changes in the carbon fluxes, but also due to changes in the signature of the carbon fluxes.
 Let us consider a single box of Volume V and inorganic C concentration DIC with an isotopic signature $\delta^{13}$C. The fluxes of preformed DIC entering and leaving the box are denoted Fp,i, with i an index for the different fluxes covering all fluxes by diffusion, advection, and convection entering or leaving the box.  Their signature is $\delta$13Cp,i.  For simplicity, we consider one flux of remineralized carbon entering the box, Fr, with the signature $\delta$13Cr,in. Mass balance is then given by:

$V * d/dt (DIC) = Sum(Fp,i) + Fr$            (2a)
$V * d/dt(DIC * \delta^{13}C) = Sum(Fp,i * \delta13Cp,i) + Fr * \delta13Cr,in$        (2b)

Subtracting steady state fluxes and considering the change ($\Delta$) over one time step of length $\Delta$t, we get  with $\Delta$(DIC) = $\Delta F/V * \Delta t$:

$\Delta$ (DIC)= $\Delta$ (DICp) + $\Delta$ (DICr)           (3a)
$\Delta$ (DIC * $\delta$13C) = $\Delta$ t/V * $\Delta$ (Sum(Fp,i * $\delta$13Cp,i) + $\Delta$ (Fr*$\delta$13Cr,in)     (3b)

Linearising (3b) and using again $\Delta$(DIC) = $\Delta F/V * \Delta t$, we get:

$\Delta$(DIC) * $\delta$13C + DIC * $\Delta$($\delta$13C) = $\Delta$(DICp) * $\delta$13Cp,in + $\Delta$(DICr)*$\delta$13Cr,in
                           $\Delta$ t/V *Sum( Fp,i * $\Delta$($\delta$13Cp,i)) + $\Delta$ t/V * Fr * $\Delta$($\delta$13Cr,in)  (4)

The first two rhs terms in (4) describe the change in isotopic mass due to the addition of carbon by the perturbed preformed and remineralized carbon fluxes. The last two rhs terms describe the change due to the change in the signature of the preformed and remineralized carbon fluxes. Equations 4 has many unknowns ($\Delta$($\delta$13Cr), $\Delta$($\delta$13Cp), $\delta$13Cr,in and $\delta$ 13Cp,in …). Thus, it seems not possible to attribute the change in $\delta$13C to preformed and remineralized components in an exact way without carrying a separate preformed d13C tracer in the model.

Perhaps it is justified to make approximations.
We may assume that
 $\Delta$(DICp) * $\delta$13Cp,in << $\Delta$ (DICr)*$\delta$13Cr,in.              (5)

This is probably o.k. as $\delta13C_{p,in}$ is close to zero permil and and d13Cr,in is about -20 permil for organic material.

It is much less clear whether also the terms with the changes in the isotopic signatures in eq. (4) can be neglected. The changes in $\delta13C$ may be small, but they are multiplied with the total carbon fluxes (Fr, Fp,i) and not just with the perturbations in the carbon fluxes. Therefore, these terms may be very significant. Nevertheless, let us assume for the moment these two terms are negligible. In this case, we get:
$\Delta(DIC) * \delta13C + DIC * \Delta(\delta13C) \sim \Delta(DICr)*\delta13Cr,in$ and the solution for $\Delta(\delta13Cr)$ is:

$\Delta(\delta13Cr) \sim \Delta(DICr)/DIC * (\delta13Cr,in - \delta13C) - \Delta(DICp)/DIC * \delta13C$       (6)

Eq. 6 is somewhat similar to the eq. (1) above and given on line 124 in the MS, when setting $\delta13Corg = \delta13Cr,in - \delta13C$. This difference in isotopic signatures of the material remineralized and of the isotopic signature of DIC should be considered. In particular, in the upper ocean $\delta13C$ of DIC is different from zero.
In addition, it seems that the parentheses are not properly set in (1) and eq. 1 should rather read $\delta13Cr = d13Corg * \Delta(DICorg)/DIC$.
Further, the second rhs term of (6) is neglected in (1). The second rhs term in eq. 6 may be small as typical source signatures are between 0 and 2 permil in the upper ocean. However, it seems easy to account for in the evaluation of $\Delta(\delta13Cr)$.

In conclusion, the calculation of the change in $\delta13C$ attributable to organic matter remineralization and to preformed fluxes must be revised. It remains the task of the authors to demonstrate that changes in the isotopic signature of the preformed and remineralized fluxes can either be safely neglected (as done to get eq. (6) or otherwise to properly account for their influence.

My recommendation is to explicitly include preformed tracers in LOVECLIM and then to repeat the simulation shown in figure 1 with the preformed tracers enabled in this model of intermediate complexity.
The simulations have been published before and the separation of d13C changes into preformed and remineralized components is the main point of this paper. Thus, this separation should be done properly to make this manuscript publishable.

2) Line 149-158: The authors separate surface ocean $\delta13C$ change into a thermodynamic equilibrium component and a residual component. I am puzzled by the interpretation offered by the authors. The authors state that the residual component mainly reflects changes in primary productivity. This is not demonstrated but only inferred from simulated changes in productivity. The balance between the input of PO4 by upwelling and consumption of PO4 by export leads to a positive PO4 anomaly in the SO surface ocean (Fig 3d). Correspondingly, the balance of upwelling and export alone leads to a negative $\delta13C$ anomaly in the SO ocean surface. It is unclear to which extent incomplete air-sea exchange contributes to this residual component.

3) The LOVECLIM simulation is forced by prescribed changes in wind stress and salt fluxes (Fig. 1a,b). This triggers a change in Southern Ocean upwelling and deep ocean ventilation (as e.g., reflected by AABW changes in Fig. 1c). One may then ask which part of the early deglacial CO2 rise may be explained by such a change in deep ocean ventilation.

   The ratio between the change in atmospheric $\delta$13C and CO2 interesting as this ratio can be directly compared with ice core data as done in previous work.

   The change in deep ocean ventilation and Southern Ocean upwelling enforced by prescribed wind stress and freshwater forcing causes $\delta$13C to decline by 0.35 and CO2 to increase by 25 ppm in the LOVECLIM simulation (Fig 1c). This yields a ratio of 7 ppm per 0.1 permil decline. (Tschumi et al. 2011) performed similar idealized simulations where Southern Ocean overturning was changed by prescribed changes in boundary conditions. They found a ratio of 13 ppm per 0.1 permil decline. The ice core data suggest an increase in CO2 of 35 ppm and a decrease in d13C of 0.3 permil during the early deglacial period. This corresponds to a ratio of 12 ppm per 0.1 permil decline. Tschumi et al.  suggested that the entire increase in CO2 of 35 ppm during the early deglacial was due to enhanced Southern Ocean upwelling. The LOVECLIM results suggest a smaller contribution of SO upwelling to the early deglacial CO2 rise. According to LOVECLIM only 25 ppm of the deglacial CO2 rise are attributable to the prominent Southern Ocean upwelling hypothesis.

   This issue should be discussed in section 4.

Minor comments:

Intro: It is suggested to change the framing of the introduction. It should be clearly pointed out that it is very well established by the modelling community and by those addressing the anthropogenic carbon perturbation that air-sea gas exchange influences d13C, but that this well-established fact is sometimes neglected in interpretation of planktonic d13C records. It would be appropriate to recall the typical equilibration time of 10 yr for d13C in the surface layer by air-sea gas exchange and the typical decadal timescale of surface-to-thermocline transport as revealed by observations of anthropogenic tracers.

L51-52: I do not understand the conclusion that atmospheric CO2 is not affected. If atmospheric CO2 (and d13C) varies/is perturbed, e.g., by outgassing in the Southern Ocean, then the CO2  perturbation will like the d13C perturbation enter the upper ocean.

L57: This is a somewhat odd description of the preformed component. The preformed component reflects the balance between all tracer sources and sinks in a surface grid cell. Upwelling and exchange with the deeper layers are generally equally important as air-sea exchange and export production. Why highlighting the terms thermodynamic equilibrium and primary productivity? Would it not be more appropriate to mention air-sea gas exchange and new or export production as well as physical tracer exchange between surface and deeper layers?

L109: "The atmosphere is held constant ..allowed to evolve freely" This text is unclear. Do you mean in the experiment "fix" atm. CO2 and d13C is kept constant and in exp. "free" CO2 and d13C evolve freely?

1.      Heimann, M., and Maier-Reimer, E.: On the relations between the oceanic uptake of CO2 and its carbon isotopes, Global Biogeochem. Cycles, 10, 89-110, 1996.

2.      Broecker, W. S., Peng, T.-H., Ostlund, G., and Stuiver, M.: The distribution of bomb radiocarbon in the ocean, Journal of Geophysical Research: Oceans, 90, 6953-6970, 10.1029/JC090iC04p06953, 1985.

3.      Eide, M., Olsen, A., Ninnemann, U. S., and Eldevik, T.: A global estimate of the full oceanic 13C Suess effect since the preindustrial, Global Biogeochemical Cycles, n/a-n/a, 10.1002/2016gb005472, 2017.

4.      Tschumi, T., Joos, F., Gehlen, M., and Heinze, C.: Deep ocean ventilation, carbon isotopes, marine sedimentation and the deglacial $CO_2$ rise, Clim. Past, 7, 771-800, 10.5194/cp-7-771-2011, 2011.

---

## Editor Comment (EC1) · Hubertus Fischer (Editor) · 2 Oct 2020

Dear authors

Your paper has been seen by two expert reviewers which both raise important questions from their individual perspective regarding the approach taken in your paper (carbon partitioning, assumptions on isotopic signatures) while at the same time acknowledging the potential impact of your paper.

The fundamental gaps in explaining your approach in the current version do not allow me to make a final decision of acceptance or rejection of this manuscript and will re-

quire another round of reviews after you carefully revised your manuscript in the light of the two reviews provided so far.

In case you decide to continue with the review process of your paper, I would ask you to detail how you will meet the points raised by the reviews in your answer concluding the discussion phase. You will then be asked to provide a completely revised version of the paper including a point-to-point reply to the review comments. This will be then the base for a re-review of the revised paper.

All the best Hubertus Fischer (editor of Climate of the Past)

---

## Author Comment (AC1) · 9 Feb 2021

Shao et al. assess the mechanisms driving changes in the stable carbon isotopic composition of the upper ocean and in atmospheric CO2 (d13CO2) during the last deglaciation, focusing on the first major decline in d13CO2 observed in Antarctic ice core records around 17 kyr before present. Based on model simulations with LOVECLIM and GENIE, the authors test two hypotheses that may explain these trends: first, the upwelling of respired carbon (with a low-d13C signature) from the deep ocean, primarily in the Southern Ocean and its advection to the global ocean via the thermocline, and subsequent equilibration with the atmosphere (bottom-up scenario); and second, the sub-surface supply of respired carbon and strong equilibration with the atmosphere in upwelling regions (causing a decrease in d13CO2), and parallel transmission of the atmospheric d13CO2 signal to the upper ocean via air-sea gas exchange (top-down scenario). Through a carbon speciation analysis, the authors find a strong influence of the top-down process on global upper-ocean d13C records (including a new one from the western equatorial Pacific), confirming important proxy-based postulations made by Lynch-Stieglitz et al. (2019).

This paper is a timely model-study on the mechanisms of global d13C records, testing (opposing) inferences on the global carbon cycle made initially by Spero and Lea (2002) and more recently by Lynch-Stieglitz et al. (2019). It therefore merits publication in Climate of the Past.

We are grateful for the positive assessment of our work.

I do, however, have difficulties to follow the argumentation of the authors in places, see why different model approaches were chosen (transient vs. equilibrium, glacial vs. interglacial boundary conditions) and whether they are appropriate for the premise of the study (in particular, their combination). The study essentially confirms the proposition of Lynch-Stieglitz et al. (2019) but I see some scope to provide novel insights that would increase the impact of the study. I elaborate on these aspects and other minor ones below. I recommend major revisions of the paper prior to publication. I also want to sincerely apologize to the authors for the delay in providing my evaluation of their manuscript. I hope that despite the delay the authors find my comments useful in improving their study.

We will revise the manuscript to better elucidate the rationale for our approach and in doing so also accommodate recommendations of Referee #2, paying particular attention to how the models and associated experiments are justified and described, how the numerical tracers are defined, as well as expand on the more novel insights that arise (including evaluation of preformed $\delta^{13}$C). This we detail in the point-by-point responses below.

We will also frame the paper much more towards the novel regenerated $\delta^{13}$C numerical tracer that we have implemented in cGENIE – this is the first time such an (explicit) analysis has been carried out to our knowledge, and enables us to shed novel insights into the different components contributing to observed $\delta^{13}$C changes as well as error inherent in previously publish approximation (from regenerated $PO_4$ to regenerated $\delta^{13}$C) approaches.

Major comments:

Preformed and remineralized speciation in Introduction: The partitioning of ocean carbon into 'preformed' and 'remineralized' is central to the authors' study, but these important terms are not properly introduced in the study. A definition of these terms in the introduction are needed, and in particular how they are defined and what processes they are influenced by in the real world and in the model world.

In the revision, we will be more expansive on the description, justification, and application of the numerical/diagnostic tracers employed in the models.

The latter I find some- what incomplete: How do kinetic equilibration effects play into the partitioning process of carbon between the atmosphere and ocean, besides thermodynamic equilibration effects and primary production? Are surface wind effects considered as drivers of air- sea gas exchange in the model? Through the impact of surface wind stress on the piston velocity or gas transfer coefficient, winds have a strong influence of air-sea gas exchange in the real world (e.g., Wanninkhof, 1992). Also note in line 59, that changes in the residence time of water parcels at the surface can also lead to preformed carbon changes, simply by varying the time available for air-sea gas exchange. This statement needs to be revised accordingly.

We are grateful to the reviewer for highlighting what was a poor descriptive effort on our part, particularly given the importance of the tracer to the study. We will substantially improve and expand on the description in the revision.

With respect to the role of winds and air-sea gas exchange (and in addition to addressing requests of Referee #2 regarding a fuller description of the global invasion of isotopic signatures from the atmosphere), we will explicitly isolate the role of changing (Southern Ocean) winds – both as influencing only air-sea gas exchange and not circulation, and in influencing only circulation and not air-sea gas exchange – in an additional series of cGENIE model experiments that pick apart the changing controls on preformed vs. regenerated $\delta^{13}$C.

Line 83. Justification is needed why the simulation LH1-SO-SHW was chosen although Menviel et al. (2018) provide a number of other simulations with increase Southern Ocean ventilation, e.g. LH1-SO.

"LH1-SO-SHW" was picked from Menviel et al, (2018) for several reasons: 1) recent ice core records also suggest enhanced SO westerly winds during Heinrich stadials (Buitzert et al., 2018); 2) "LH1-SO-SHW" matches some of the important observations (e.g. ice core record of atmospheric $pCO_2$ and $\delta^{13}CO_2$) better than the other scenarios presented in Menviel et al.,(2018); 3) the stronger SO windstress in "LH1-SO-SHW" leads to an increased transport of AAIW to lower latitudes, which could have impacted the intermediate depths of the global ocean, including the site of our new benthic $\delta^{13}$C record.

Offline calculations of carbon species in LOVECLIM: I find it striking that the authors' "approach requires accurate representation of the preformed and remineralized components" (line 62), but that the LOVECLIM model does not simulate them explicitly. The authors need to discuss what types of errors might affect their offline calculation based on the LOVECLIM and how large these errors might be. For instance, why does AOU overestimate true oxygen utilization? (line 182).

This goes to the heart of our '2-model' methodology (also see replies to Referee #2), in that we are re-analyzing an existing model experiment (LOVECLIM 'LH1-SO-SHW') and that the particular published experiments we are interested in lack the specific (and unique) numerical tracer we need. For this reason, we employed the 'cGENIE' Earth system model of intermediate complexity to explicitly evaluate metrics derived from the LOVECLIM model experiment – AOU to regenerated phosphate and hence to regenerated $\delta^{13}C$. Furthermore, rather than evaluate derived metrics such as AOU in the context of the modern (preindustrial) state, we will conduct additional experiments employing glacial-like boundary conditions in cGENIE and carry out the evaluation in that context. This will all be significantly expanded upon in the revised manuscript, including discussion of errors inherent in the approximations.

With respect to the Referee's specific question – it is well known that AOU likely over estimates the true oxygen utilization, and thus $DIC_{org}$, particularly in water masses formed in high latitudes (Bernardello et al., 2014; Ito et al., 2004; Khatiwala et al., 2019). To this, we will provide illustrative maps of the AOU error to give the reader a better sense of where (and why) the AOU approximation breaks down. We will present a similar analysis for the step to regenerated $\delta^{13}C$.

I find the sensitivity experiments made in cGENIE to alleviate the problems associated with the necessity of an offline calculation not convincing, because the experimental setup, forcing and boundary conditions are very different. This leads to my next point of criticism.

Comparability and suitability of LOVECLIM and cGENIE simulations: How do the cGENIE and LOVECLIM simulations support each other, when they are so different? Is it correct that wind changes are not considered in the cGENIE simulation (which they are in the LOVECLIM simulation)? If correct, this should be clearly stated. In that case, would this call for the use of LH1-SO instead of LH1-SO-SHW?

Firstly, we agree that the 2-model methodology was not made clear from the outset. We propose an extensive revision of the text that separates out the cGENIE-based assessment of how (and how reliably) regenerated $\delta^{13}C$ can be estimated in model (in turn based on AOU) before moving onto the analysis of the LOVECLIM experiment. We will include explicit graphical illustration and discussion (also addressing comments by Referee #2) that supports what will be a much more transparent and logical methodology.

Secondly, we agree with the reviewer that since we employ cGENIE to evaluate the method we use to attribute the isotope changes simulated in LOVECLIM, that the experimental design for cGENIE should be as close as possible to that of LOVECLIM. Hence, for the revision, we will carry out a revised series of tracer diagnostics and analysis using cGENIE simulations run under recently published and more 'glacial-like' conditions that account for a different planetary albedo

due to expanded continental ice sheets as well as the radiative forcing from the lower glacial greenhouse gas concentration (Rae et al., 2020). To better compare with LH1-SO-SHW, we will also include transient varying wind stress forcing over the Southern Ocean in the cGENIE experiments, in addition to the salt/freshwater flux that is already applied in the original simulations.

How preformed nutrients or carbon are simulated in cGENIE is unclear, in particular given the statement in line 98 to 99. If preformed tracer values are reset to the full tracer value (what is this?) at each model step, does this skew the outcome to a dominance of preformed changes? I believe some more explanation is required here, as this suggests that all water masses leaving the surface ocean, e.g. in the Southern Ocean, have no remineralized tracer component.

The cGENIE model still carries a DIC (and $^{13}C_{DIC}$) tracer, which when leaving the surface can accumulate remineralized (regenerated) DIC (and $^{13}C_{DIC}$). In addition to this standard tracer, we include a pre-formed DIC (and $^{13}C_{DIC}$) tracer that indeed does leave the ocean surface initially with no regenerated component and only accumulates regenerated DIC (and $^{13}C_{DIC}$) subsequently. We will make this much clearer in the revised text.

Relative contributions of top-down and bottom-up processes: The authors suggest that air-sea gas equilibration leaves a strong imprint on upper-ocean d13C records, while also acknowledging that bottom-up processes cannot be neglected, more so in some regions over others (e.g., line 174-179). However, the authors focus a lot on the top-down process, while in my view they would be in the position of *quantifying* what the relative contributions of these different processes in *different regions* are (and provide a global map accordingly). This would significantly increase the impact and value of the study, in particular for those researchers working with proxy data. I hence encourage the authors to consider performing these analyses. The study should also better highlight the finding that upper-ocean d13C are ultimately affected by both (top-down and bottom-up) processes but with strongly varying proportions in different regions.

We thank the reviewer for the suggestion. Indeed, showing a relative contribution of net $\delta^{13}C$ anomaly of preformed versus regenerated component would be very helpful for paleo tracer community. However, such a quantitative 'map' for the early deglaciation may very much depend on the models used, boundary conditions and forcing applied. This can already be seen by comparing the LOVECLIM and cGENIE simulations provided in the present study. Nonetheless, based on the zonal sections of the Pacific that show how the net change in $\Delta\delta^{13}C$ breaks down into preformed and regenerated components, we can make some useful qualitative statements such as: "$\Delta\delta^{13}C_{pref}$ dominates the upper 1000m and could account for a 0.3-0.4‰ decline in marine planktic records during the early deglaciation, whereas $\Delta\delta^{13}C_{reg}$ becomes increasingly important at deeper depth" and which we will expand on in the revision. We will also provide comparable zonal sections for the Atlantic and Indian Ocean basins and thereby provide something equivalent to a 'map' (one broken down into zonal sections).

Focus on initial deglacial d13CO2 decline: It is confusing that in places the entirety of the deglacial d13CO2 is discussed although boundary conditions and driving mecha- nisms might differ throughout the deglaciation (e.g. 162-164). I recommend to remove these and instead exclusively focus on the early deglacial d13CO2 change.

We will now focus on the early deglacial part of the record as suggested by the reviewer.

The same (somewhat) applies to the centennial change in pCO2 around 16.2 kyr before present (e.g., 206-208).

Lines 206-208 refer to $\delta^{13}CO_2$ rather than atmospheric $CO_2$ at 16.2ka. We argue that the centennial negative $\delta^{13}CO_2$ excursion documented by the Taylor glacial record is part of the early deglacial $\delta^{13}CO_2$ change.  If the atmospheric bridge is really efficient as we propose, this rapid negative $\delta^{13}CO_2$ excursion should have had a strong influence on the global upper ocean, although a centennial marine signal is not likely to be captured by most of the sedimentary records. The LOVECLIM simulation illustrates nicely that such a centennial marine signal can be visible in the simulated global upper ocean water mass, supporting a highly efficient atmospheric bridge in transporting $\delta^{13}C$ anomaly. Thus, we would like to keep the discussion about the centennial change in $\delta^{13}CO_2$ around 16.2 ka in the revision.

Representation of foraminiferal d13C of true DIC d13C changes: It might be worthwhile to highlight in the manuscript that the one-to-one representation of seawater DIC d13C changes based on foraminiferal d13C is imperfect, more so for planktonics than for benthics (e.g., Bemis et al., 2000; Schmittner et al., 2017). It might be hence useful to clarify whether the trends and/or the magnitude of benthic d13C change resembles atmospheric d13C changes, e.g., in line 261, and whether both can be linked without reservations.

We thank the reviewer for this helpful suggestion. We will add some relevant descriptions in the introduction so that the readers are aware of the issues related to foraminiferal $\delta^{13}C$ records.

The paragraph will be changed along the following lines of:

"Here we term this scenario 'bottom up' transport. In this scenario, the upper ocean at lower latitudes acts as a conduit for $^{13}C$-depleted carbon to enter the atmosphere. As a result, benthic foraminifera upper intermediate depths of low latitude oceans should have also recorded such an early deglacial $\delta^{13}C$ decline. We are aware that benthic (Schmittner et al., 2017) and planktic (e.g. Bemis 2000) $\delta^{13}C$ can be complicated by temperature (planktic) and carbonate ion changes (both). Thus foraminiferal $\delta^{13}C$ changes at different parts of the upper ocean may not totally reflect seawater DIC $\delta^{13}C$ changes. Nonetheless, foraminiferal $\delta^{13}C$ changes (especially benthic foraminifera) are highly correlated with seawater DIC $\delta^{13}C$ changes."

Minor comments:

Line 23. Specify the depths that relate to "from depths that are potentially affected by the atmosphere".

We will specify the depths (i.e. upper 1000m) as suggested.

Line 28. I find that the statement "The mechanisms and the chain of events that were responsible for this pCO2 are not well understood" neglects a large body of literature, a large number of existing hypotheses and a wealth of proxy-data in support of some of these. I recommend some more nuance and adjustments to reflect this. E.g. "De- spite xxx, the mechanisms ..." or "Although the leading hypothesis for millennial- and centennial-scale pCO2 rise was suggested to be xxx, the chain of events ..."

The paragraph will be changed along the following lines:

"Atmospheric $pCO_2$ increased by 80-100 ppm from the last glacial maximum (LGM) to the Holocene (Marcott et al., 2014; Monnin et al., 2001). During the initial ~35ppm rise in $pCO_2$ rise between 17.2 to 15 ka, ice core records have documented a 0.3‰ decrease in atmospheric $\delta^{13}C$ (Bauska et al., 2016; Schmitt et al., 2012). This millennial-scale trend was punctuated by a rapid 12ppm $pCO_2$ increase between 16.3-16.1 ka (Marcott et al., 2014) and a 0.2‰ decrease in $\delta^{13}CO_2$ (Bauska et al., 2016). Leading hypotheses that have been proposed to explain the early deglacial carbon cycle perturbation includes increased Southern Ocean ventilation (e.g. Skinner et al., 2010, Burke et al., 2012), poleward shift/enhanced Southern Hemisphere westerlies (Toggweiler et al., 2006, Anderson et al., 2009, Menviel et al., 2018) and reduced iron fertilization (Martínez-García et al., 2014). However, the chain of events is not well understood."

Line 44. I do not think that a clear lead of a d13CO2 decline can be or was documented. I hence recommend removing "initially occurring in the atmosphere"

This statement will be removed.

Line 51. The statement "and the subsequent d13C decline . . ." needs to be revised as it is confusing. How can a d13C decline contribute to pCO2 variability? I recommend changing it to "is a reflection of the evasion of oceanic carbon to the atmosphere, contributing to . . ."

This statement will be removed.

Line 63. Specify what components.

Errors in estimated regenerated DIC will also affect preformed component as the latter is calculated as the difference between simulated DIC and estimated regenerated DIC. We will explicitly clarify this in the revision.

Line 70. "To our knowledge, the origin .." this sentence is confusing and seems out of place. Please revise.

The text will be removed.

Line 72. It is entirely unclear at this stage why a new benthic d13C record has been obtained. This sentence should be moved or the premise of these analyses should be introduced.

One of the main findings of our study is that this fast equilibrium $\delta^{13}$C route through the atmospheric bridge compared to ocean transport actually affects not only the top layers in the ocean (i.e. where planktic foraminifera live), but also the water column down to perhaps 1000m.

The motivation for presenting a new benthic $\delta^{13}$C record from upper intermediate Pacific will be clearly described in the Introduction. We will also improve the structure of the paper earlier on to better justify and explain how the new data fits in with the overall methodology.

Line 87. Insufficiencies of the models in representing sub-grid processes are unquestionable. This statement should not be phrased as if they were not.

Our apologies – this is not what we intended to say. The sentence will be changed along the following lines:

"Due to its relatively coarse resolution, the model could mis-represent the high southern latitude atmospheric or oceanic response to a weaker NADW. Enhanced AABW could have occurred due to a strengthening of the SH westerlies, changes in buoyancy forcing at the surface of the Southern Ocean, opening of polynyas, or sub-grid processes."

Line 108. It is entirely unclear why the forcing is limited to the Pacific sector of the Southern Ocean. Please specify. Here for consistency, I recommend changes a similar forcing to Menviel et al., (2018).

In a revised series of experiments, we have now applied salt flux forcing to the entire SO in cGENIE experiments and hence to better align with the LOVECLIM experiment.

Line 120. A full sentence is needed here. Also, DICorg is depleted in 13C not d13C. Line 121. Budget of what?

We will revise the paragraph to address these points.

Line 123: (Dd13Creg) instead of (Dd13C)

Yes, our mistake (which will be corrected).

Line 124: Is d13Corg defined or simulated? Is DIC =DIC total, i.e. reg + preformed? How is 12Corg defined?

DIC=DICtotal=DICreg+DICpref.

In the revision, we will stick to 'DIC' and not additionally use 'DICtotal' to avoid confusion.

In the original submission, $^{12}C_{org}$ was defined as -21‰ that matches the observed modern global mean $\delta^{13}$C of POC (Goericke & Fry 1994). However, depending on the choice of parameterization, the modelled $\delta^{13}$C of POC can be different from -21‰ (Dentith et al., 2020). In

the revision, to be self-consistent, $^{12}C_{org}$ will be defined as the simulated global mean $\delta^{13}C$ of POC in each model. We thank the reviewer for catching this.

Line 129. 2 and 5 mg CaCO3.

Fixed.

Line 131-132. What suggests that there is no evidence for invariable surface ocean reservoir age changes over the deglaciation? It is not enough to say that. I believe it has to be justified. Also Figure 4 shows a marked lag between the onset of d13C decline in the GeoB17402 and in atmospheric d13CO2. Is this real or an artifact of the age model (i.e., variable reservoir ages?)? I am surprised that there is no men- tion/discussion of this lag in the study.

We now use the new Marine20 calibration curve that incorporates potential reservoir changes to update our age model. However, the lag the reviewer was referring to still exists and we attribute it to a relatively large age model uncertainty below 154cm (median age ~16.2yr), up to 1-2 kyr (2SD)

Line 133. Remove "Once the calendar ages were established the results were plotted vs depth."

Removed.

Line 140. Remove "will be archived in Pangaea" and add URL to appropriate section Data availability.

We will obtain an URL, which will then be added into our revised manuscript.

We will also make the cGENIE experiment configuration files (and instructions for running the experiments) available on GitHub and generate a DOI for this.

Line 142-143. Remove "Below.. " I don't find this helpful here, and the structure of the manuscript can be reflected in the headings.

This sentence will be removed.

Line 149. Which model?

The LOVECLIM model. We will better clarify this in the text.

Line 152-154. I am surprise to see a discussion of entirely new carbon species/terms (Dd13Cthermo and Dd13Cres), which haven't been introduced or mentioned earlier. They need to be properly introduced, otherwise this analysis is entirely confusing, and not very helpful for the reader. They also appear not to be of relevance throughout the remainder of the manuscript, which somewhat questions whether this analysis is needed. It is difficult to follow the statements in the following lines 154- 157: What is meant here with Dd13C? What does the preformed signal reflect? Dd13Cthermo? Please clarify.

This section will be removed for clarity. We were over complicating things unnecessarily with 'Dd13Cthermo and Dd13Cres'.

Line 165. It should be pointed out clearly what observations lead to this major finding.

The $\delta^{13}C$ anomaly in the upper 1000m of the ocean is dominated by the preformed $\delta^{13}C$ signal leads to this finding. We will be more specific in the revision. As we will regarding the novelty of the creation of an explicit preformed $\delta^{13}C$ tracer in an Earth system model.

Line 172. "evolution" instead of "pathway"

We will change the wording as suggested.

Line 188. The d13C decline in the upper 1000 m (where? Does Figure 6 show a global ocean mean?) is also dominated by the preformed signal (everywhere?). Also some more help and explanation with regards to Figure 6 is needed, as it shows four panels.

Figure 6 are the zonal mean Pacific plots simulated by cGENIE, we will make it clear in the caption and in the associated main text.

Line 215-217: Reference to a figure is required.

We will add this.

Line 277-280: Please specify what time interval you refer to here. This also seems like an add-on that is not properly analyzed, and I hence wonder how useful this is. The authors would be in the position to test the different hypotheses of why the Atlantic and Pacific anomalies are so different, but that is entirely glossed over at this stage.

17.2-15ka is the interval. This paragraph is really about how benthic $\delta^{13}C$ records from 2000m of the South Atlantic can be re-interpreted with the insight from the transient simulation. So the Pacific-Atlantic difference is indeed an unnecessary add-on. We will remove the vague discussion in the revision.

References:

Dentith, J. E., Ivanovic, R. F., Gregoire, L. J., Tindall, J. C. and Robinson, L. F.: Simulating stable carbon isotopes in the ocean component of the FAMOUS general circulation model with MOSES1 (XOAVI), Geoscientific Model Development, 13(8), 3529–3552, https://doi.org/10.5194/gmd-13-3529-2020, 2020.

Goericke, R. and Fry, B.: Variations of marine plankton $\delta^{13}$C with latitude, temperature, and dissolved $CO_2$ in the world ocean, Global Biogeochemical Cycles, 8(1), 85–90, https://doi.org/10.1029/93GB03272, 1994.

Rae, J. W. B., Gray, W. R., Wills, R. C. J., Eisenman, I., Fitzhugh, B., Fotheringham, M., Littley, E. F. M., Rafter, P. A., Rees-Owen, R., Ridgwell, A., Taylor, B. and Burke, A.: Overturning circulation, nutrient limitation, and warming in the Glacial North Pacific, Science Advances, 6(50), eabd1654, https://doi.org/10.1126/sciadv.abd1654, 2020.

---

## Author Comment (AC2) · 9 Feb 2021

Fortunat Joos

The paper forcefully demonstrates that air-sea gas exchange effectively acts to couple atmospheric and upper ocean d13C. This point is, - from a modelling point of view and for all those monitoring the penetration of the anthropogenic perturbation into the ocean - rather trivial and not new. The timescale to bring the surface layer in equilibrium with a d13C perturbation in the atmosphere by air-sea gas exchange is of order 10 years as outlined by Broecker, Peng and others. Numerous measurements of CFCs, bomb-produced radiocarbon, DIC, and notably of d13C demonstrate that the atmospheric perturbation in these tracers is communicated by air-sea gas exchange to the surface layer and by surface-to-deep exchange to deeper layers within years to decades (e.g., (Heimann and Maier-Reimer 1996;Broecker et al. 1985;Eide et al. 2017). Thus it is clear from an observational as well as from a modelling point that air-sea gas exchange is important and needs to be considered when addressing carbon isotopes. Unfortunately, the role of air-sea exchange is sometimes neglected in the interpretation of marine planktonic d13C records. It may therefore be appropriate to recall this point for the paleoceanographic tracer community.

We thank the reviewer for his very thorough and helpful review. We agree that the principle behind a dominant rule of air-sea gas exchange on surface $\delta^{13}C$ is quite well established in theory and is 'well known' in the modeling community (e.g. Schmittner et al., 2013). We are very happy to revise the manuscript to include these points and better highlight the coupling between ocean and atmosphere. However, in the specific hypotheses we address, the situation is subtly different – although eventually the signature of isotopically depleted carbon release must reach the atmosphere and hence be re-transmitted to the entire global surface ocean (via air-sea gas exchange), the question is whether the observed $\delta^{13}C$ decline in specific planktic and shallow/intermediate depth benthic records reflect the 'end point' (following ocean invasion of the signature globally from the atmosphere), or whether they reflect a location on the pathway of carbon release from the ocean (and hence prior to invasion to the atmosphere and global surface re-equilibrium). We admit this was not fully clear in the original text and will discuss the competing hypotheses (plus sequence of events, and the inevitable role of air-sea gas exchange) much more clear.

Interesting is that the authors offer a quantification of the influence of preformed versus remineralized changes in d13C. However, the method applied to separate changes in d13C into the contribution from preformed sources and sources from biogenic particles is unclear and may not be appropriate (see below).

In the cGENIE model the numerical tracer is exact by definition, but we agree that there are a number of caveats that we did not discuss when deriving regenerated $\delta^{13}C$ from other modelled metrics (as we do in LOVECLIM). We address this in response to the detailed Referee comments below.

Another interesting point, which deserves a bit more discussion, is the information on the change in d13C versus the change in atmospheric CO2 in response to an increase in deep ocean

ventilation (AABW, AAIW) forced by prescribed changes in salt and Southern Ocean wind stress.

I recommend major revisions. Specific comments:

1) Attribution of the d13C changes.

Section 2.3 The separation into preformed and remineralized d13C from the model output of DIC and d13C appears problematic. The assumption and simplifications of the approach are not explained to the readers

We will fully rectify this.

 It would be preferable to simulate the preformed tracers online.

We do in cGENIE. But the historical/published nature of the LOVECLIM experiments means that we are unable to simulate the preformed tracers online for LOVECLIM (hence the use of cGENIE to elucidate the errors in the approximation we then must use).

1. a)  The authors use the equations given in line 124 to estimate the change in $\delta$13C due to a change in remineralization. However, the equation is unclear. A new term is used in this equation: "12Corg". I guess 12Corg should read DICorg, the amount of remineralized carbon. Then the equation given by the authors reads:

   $\delta$13Cr=d13Corg * $\Delta$(DICorg /DIC) (1)

   We admit some sloppiness with the notation, which we will correct and much better clarify.

2. b)  How is $\delta^{13}$Corg computed? I get the impression that a constant $\delta^{13}$C signature of organic carbon ($\delta^{13}$Corg) is assumed in the approach applied to distinguish the preformed and remineralized components in LOVECLIM.

   As per earlier comments (and also subsequent replies) – we will extensively revise the text to more logically and explicitly detail: the role of cGENIE vs. LOVECLIM, how the preformed tracers are simulated in cGENIE, how respired $\delta^{13}$C is estimated in LOVECLIM and what the error (based on cGENIE analysis) are.

c) Any perturbation in surface $\delta^{13}$C is also transferred to newly formed organic matter and CaCO3 and finally to remineralized carbon. In addition, changes in surface CO2 affect the fractionation factor for organic matter formation and thereby again d13Corg. Further changes in

surface d13C also affect changes in the signature of preformed fluxes. It appears that changes in the isotopic signature of preformed and remineralized carbon fluxes are neglected. This seems an oversimplification.

We agree, and it was an omission not to discuss this in the original manuscript. (We note that the cGENIE preformed tracer analysis encapsulates *all* errors, but these were not broken down as highlighted by the Referee.)

d) The equation on line 124 used to compute the change in remineralized $\delta13C$ needs to be properly derived. The mass balance should be considered in the separation of the different components. I distinguish preformed DIC (DICp; index p) and remineralized DIC (DICr; index r) and related fluxes (Fp, Fr).

Changes in $\delta13C$ can arise due to changes in the carbon fluxes, but also due to changes in the signature of the carbon fluxes.

Let us consider a single box of Volume V and inorganic C concentration DIC with an isotopic signature $\delta^{13}C$. The fluxes of preformed DIC entering and leaving the box are denoted Fp,i, with i an index for the different fluxes covering all fluxes by diffusion, advection, and convection entering or leaving the box. Their signature is $\delta13Cp,i$. For simplicity, we consider one flux of remineralized carbon entering the box, Fr, with the signature $\delta13Cr,in$. Mass balance is then given by:

$V * d/dt (DIC) = Sum(Fp,i) + Fr$ (2a) $V * d/dt(DIC * \delta^{13}C) = Sum(Fp,i * \delta13Cp,i) + Fr * \delta13Cr,in$ (2b)

Subtracting steady state fluxes and considering the change ($\Delta$) over one time step of length $\Delta t$, we get with $\Delta(DIC) = \Delta F/V * \Delta t$:

$\Delta (DIC) = \Delta (DICp) + \Delta (DICr)$ (3a) $\Delta (DIC * \delta13C) = \Delta t/V * \Delta (Sum(Fp,i * \delta13Cp,i) + \Delta (Fr*\delta13Cr,in)$ (3b)

Linearising (3b) and using again $\Delta(DIC) = \Delta F/V * \Delta t$, we get:
$\Delta(DIC) * \delta13C + DIC * \Delta(\delta13C) = \Delta(DICp) * \delta13Cp,in + \Delta(DICr)*\delta13Cr,in$

$\Delta t/V *Sum( Fp,i * \Delta(\delta13Cp,i)) + \Delta t/V * Fr * \Delta(\delta13Cr,in)$ (4)

The first two rhs terms in (4) describe the change in isotopic mass due to the addition of carbon by the perturbed preformed and remineralized carbon fluxes. The last two rhs terms describe the change due to the change in the signature of the preformed and remineralized carbon fluxes. Equations 4 has many unknowns ($\Delta(\delta13Cr), \Delta(\delta13Cp), \delta13Cr,in$ and $\delta 13Cp,in$ ...). Thus, it seems not possible to attribute the change in $\delta13C$ to preformed and remineralized components in an exact way without carrying a separate preformed d13C tracer in the model.

(Indeed, hence the inclusion of the new preformed $\delta^{13}C$ tracer in cGENIE.)

Perhaps it is justified to make approximations. We may assume that

$\Delta$(DICp) * $\delta$13Cp,in << $\Delta$ (DICr)*$\delta$13Cr,in. (5)

This is probably o.k. as $\delta$13Cp,in is close to zero permil and and d13Cr,in is about -20 permil for organic material.

It is much less clear whether also the terms with the changes in the isotopic signatures in eq. (4) can be neglected. The changes in $\delta$13C may be small, but they are multiplied with the total carbon fluxes (Fr, Fp,i) and not just with the perturbations in the carbon fluxes. Therefore, these terms may be very significant. Nevertheless, let us assume for the moment these two terms are negligible. In this case, we get:

$\Delta$(DIC) * $\delta$13C + DIC * $\Delta$($\delta$13C) ~ $\Delta$(DICr)*$\delta$13Cr,in and the solution for $\Delta$($\delta$13Cr) is:

$\Delta$ ($\delta$13Cr) ~ $\Delta$(DICr)/DIC * ($\delta$13Cr,in – $\delta$13C) - $\Delta$(DICp)/DIC * $\delta$13C (6)

Eq. 6 is somewhat similar to the eq. (1) above and given on line 124 in the MS, when setting $\delta$13Corg= $\delta$ 13Cr,in- $\delta$13C. This difference in isotopic signatures of the material remineralized and of the isotopic signature of DIC should be considered. In particular, in the upper ocean $\delta$13C of DIC is different from zero.
In addition, it seems that the parentheses are not properly set in (1) and eq. 1 should rather read $\delta$13Cr=d13Corg * $\Delta$(DICorg) /DIC.
Further, the second rhs term of (6) is neglected in (1). The second rhs term in eq. 6 may be small as typical source signatures are between 0 and 2 permil in the upper ocean. However, it seems easy to account for in the evaluation of $\Delta$($\delta$13Cr).

In conclusion, the calculation of the change in $\delta$13C attributable to organic matter remineralization and to preformed fluxes must be revised. It remains the task of the authors to demonstrate that changes in the isotopic signature of the preformed and remineralized fluxes can either be safely neglected (as done to get eq. (6) or otherwise to properly account for their influence.

We intend to address this via a full attribution analysis of the factors influencing $\delta^{13}$C in the ocean. Using cGENIE, we will start by elucidating the error terms involved in making the step (employed in LOVECLIM): AOU → regenerated PO$_4$, and in the context not only of pre-industrial steady-state conditions, but under transient deglacial-like boundary condition changes. We will then do similarly (using cGENIE) for the step: regenerated PO$_4$ → regenerated $\delta^{13}$C. To fully break down the error terms (as outlined above by the Refree), we will carry out a series of cGENIE experiments in which we: (a) we fix the $^{13}$C fractionation into organic matter (i.e. making it independent of changes in [$CO_{2(aq)}$]) and (b) run with and without fixed atmospheric composition. Together, we should be able to explicitly elucidate all the contributions to changing ocean $\delta^{13}$C and hence in respect in reconstructing regenerated $\delta^{13}$C from AOU in LOVECLIM, not only what the net error is (which we included in the submitted manuscript), where how large and from where the contributing terms arise.

My recommendation is to explicitly include preformed tracers in LOVECLIM and then to repeat the simulation shown in figure 1 with the preformed tracers enabled in this model of intermediate complexity.

In an ideal world, yes. However, implementing new tracers in LOVECLIM and then repeating a previously published experiment is not practical and would require several months of work and run-time. We see our approach as analogous to the CMIP/PMIP series of model inter-comparison experiments, where published experiments are subsequently 'mined' and reanalyzed (and typically without re-coding and re-running). Our methodology is somewhat aligned with this workflow.

The simulations have been published before and the separation of d13C changes into preformed and remineralized components is the main point of this paper. Thus, this separation should be done properly to make this manuscript publishable.

Accepted. See above for the additional cGENIE modelling that we propose to fully elucidate the different sources of error involved in approximating regenerated $\delta^{13}C$. The advantage of this approach over a single model run presenting only an explicit (numerical tracer) preformed $\delta^{13}C$ tracer based analysis is that it gives us the chance to evaluate how different processes control $\delta^{13}C$ distributions and changes in the ocean (as outlined by the Referee above) together with the uncertainties involved in published approaches of approximating based on AOU or preformed phosphate tracers.

As to the specific mass balance calculations, we will modify the relevant method section along the lines of:

"Our formulation is based on the following mass balance:

$$\delta^{13}C * DIC = \delta^{13}C_{pref} * DIC_{pref} + \delta^{13}C_{reg} * DIC_{reg} \quad (1)$$

$\delta^{13}C$ anomaly can be expressed as:

$$\Delta\delta^{13}C = \Delta(\delta^{13}C_{pref} * DIC_{pref} / DIC) + \Delta(\delta^{13}C_{reg} * DIC_{reg} / DIC) \quad (2)$$

The first and second term on the RHS represents the $\delta^{13}C$ anomaly that due to changes in the preformed and regenerated component, respectively.

Since the regenerated component is dominated by organic carbon and there is no $^{13}C$ fractionation during $CaCO_3$ formation in the model, the second term on the RHS $\sim \Delta(\delta^{13}C_{org} * DIC_{org} / DIC)$. We use AOU to estimate dissolved organic carbon and its contribution to the $\delta^{13}C$ anomaly: $\Delta(\delta^{13}C_{org} * DIC_{org} / DIC) = \Delta(\delta^{13}C_{org} * AOU * R_{c:-o2})$, where $\delta^{13}C_{org}$ is estimated by the global mean $\delta^{13}C$ of POC ($\sim$ -31‰) as simulated in LOVECLIM, $R_{c:-o2}$ = 117:-170.

This leads to:

$$\Delta\delta^{13}C = \Delta(\delta^{13}C_{pref} * DIC_{pref} / DIC) + \Delta(\delta^{13}C_{org} * AOU * R_{c:-o2})\ (3)$$

The anomaly is defined as the difference between 15 and 17.2 ka, equation (3) thus expands as:

$$\delta^{13}C^{15ka} - \delta^{13}C^{17.2ka} = \delta^{13}C_{pref}{}^{15ka} * DIC_{pref}{}^{15ka} / DIC^{15ka} - \delta^{13}C_{pref}{}^{17.2ka} * DIC_{pref}{}^{17.2ka} / DIC^{17.2ka} +$$
$$\delta^{13}C_{org}{}^{15ka} * AOU^{15ka} * R_{c:-o2} / DIC^{15ka} - \delta^{13}C_{org}{}^{17.2ka} * AOU^{17.2ka} * R_{c:-o2} / DIC^{17.2ka} \quad (4)$$

It is well known that AOU likely overestimates the true oxygen utilization, and thus DIC$_{org}$, particularly in water masses formed in high latitudes (Bernardello et al., 2014; Ito et al., 2004; Khatiwala et al., 2019). However, to what extent these biases will affect the relative contribution of preformed and regenerated carbon pool on $\delta^{13}C$ anomaly in a carbon cycle perturbation event has never been evaluated. To validate the results we obtained from LOVECLIM, we conducted a benchmark test with another Earth System model – cGENIE."

2) Line 149-158: The authors separate surface ocean δ13C change into a thermodynamic equilibrium component and a residual component. I am puzzled by the interpretation offered by the authors. The authors state that the residual component mainly reflects changes in primary productivity. This is not demonstrated but only inferred from simulated changes in productivity. The balance between the input of PO4 by upwelling and consumption of PO4 by export leads to a positive PO4 anomaly in the SO surface ocean (Fig 3d). Correspondingly, the balance of upwelling and export alone leads to a negative δ13C anomaly in the SO ocean surface. It is unclear to which extent incomplete air-sea exchange contributes to this residual component.

It's true that in the model, even though the surface productivity increased, the overall efficiency of the biological pump decreases when deep ocean overturning rate increases. It's also true that incomplete air-sea exchange is somewhat ignored in this separation. To focus on the main point of this paper, we will remove this part in the revision. With hindsight, this decomposition was one step in $\delta^{13}C$ attribution too far and wholly unnecessary (and confusing as also remarked upon by Referee #1).

3) The LOVECLIM simulation is forced by prescribed changes in wind stress and salt fluxes (Fig. 1a,b). This triggers a change in Southern Ocean upwelling and deep ocean ventilation (as e.g., reflected by AABW changes in Fig. 1c). One may then ask which part of the early deglacial CO2 rise may be explained by such a change in deep ocean ventilation.

The ratio between the change in atmospheric δ13C and CO2 interesting as this ratio can be directly compared with ice core data as done in previous work.
The change in deep ocean ventilation and Southern Ocean upwelling enforced by prescribed wind stress and freshwater forcing causes δ13C to decline by 0.35 and CO2 to increase by 25 ppm in the LOVECLIM simulation (Fig 1c). This yields a ratio of 7 ppm per 0.1 permil decline. (Tschumi et al. 2011) performed similar idealized simulations where Southern Ocean overturning was changed by prescribed changes in boundary conditions. They found a ratio of 13 ppm per 0.1 permil decline. The ice core data suggest an increase in CO2 of 35 ppm and a decrease in d13C of 0.3 permil during the early deglacial period. This corresponds to a ratio of 12 ppm per 0.1 permil decline. Tschumi et al. suggested that the entire increase in CO2 of 35 ppm during the early deglacial was due to enhanced Southern Ocean upwelling. The

LOVECLIM results suggest a smaller contribution of SO upwelling to the early deglacial CO2 rise. According to LOVECLIM only 25 ppm of the deglacial CO2 rise are attributable to the prominent Southern Ocean upwelling hypothesis.

This issue should be discussed in section 4.

The different $\Delta pCO_2/\Delta\delta^{13}CO_2$ sensitivity the reviewer is referring to can be mainly explained by different initial conditions in the two studies - Tschumi et al., 2011 applied pre-industrial conditions while the transient LOVECLIM simulation analyzed in this study started from a LGM state, that was benchmarked against benthic $\delta^{13}C$ data. Deep ocean $\delta^{13}C$ is ~0.6‰ lighter at the LGM than the Holocene (Peterson et al., 2014). Therefore, for the same magnitude of atmospheric $pCO_2$ increase through enhanced SO upwelling, $\delta^{13}CO_2$ decline in LOVECLIM is larger. Appropriate initial conditions are necessary to investigate the carbon cycle perturbation that led to a negative excursion in $\delta^{13}CO_2$ during the early deglaciation.

Menviel et al. 2015 present the atmospheric $pCO_2$, $\delta^{13}CO_2$ and oceanic d13C responses to changes in oceanic circulation in LOVECLIM and the Bern3D under pre-industrial conditions. As seen in their figure 4, for changes in Southern Ocean buoyancy forcings, $\Delta pCO_2/\Delta\delta^{13}CO_2$ ratio is +16ppm per 0.1‰ decline in LOVECLIM and +8.3ppm per 0.1‰ decline in Bern3D; for changes in SH westerlies, $\Delta pCO_2/\Delta\delta^{13}CO_2$ ratio is +10ppm per 0.1‰ decline in LOVECLIM.

As noted by the reviewer (and unfortunately not included in Menviel, et al. 2015), changes in SH westerlies lead to a 12 ppm $CO_2$ increase for a 0.1‰ $\delta^{13}CO_2$ decrease in the Bern3D as per Tschumi et al., 2011. The (slight) difference in sensitivity between the models could come from differences in the initial oceanic $\delta^{13}C$ distributions with a negative $\delta^{13}C$ bias in the equatorial regions at intermediate depth in LOVECLIM (see figure 3 of Menviel et al., 2015).

We think changes in atmospheric pCO2 and $\delta^{13}CO_2$ during the early part of the deglaciation represent the integrated signal of different processes: a significant AMOC weakening, an increase in SO ventilation, a decrease in SO sea-ice cover, an increase in globally averaged SST, and changes in the terrestrial biosphere. While idealized simulations of changes in SH westerlies provide information on the probable $\Delta pCO_2/\Delta\delta^{13}CO_2$ change, the actual integrated response is more complex.

Although this topic is of interest, it is more related to published work rather than the main point of this paper - $\delta^{13}CO_2$ decline during the early deglaciation can account for most of the marine planktic and shallow/intermediate benthic $\delta^{13}C$ decline in the global ocean. Thus we will add some elements of the above discussions to the Introduction rather than adding a new discussion section.

Minor comments:

Intro: It is suggested to change the framing of the introduction. It should be clearly pointed out that it is very well established by the modelling community and by those addressing the anthropogenic carbon perturbation that air-sea gas exchange influences d13C, but that this well-established fact is sometimes neglected in interpretation of planktonic d13C records. It would be

appropriate to recall the typical equilibration time of 10 yr for d13C in the surface layer by air-sea gas exchange and the typical decadal timescale of surface-to-thermocline transport as revealed by observations of anthropogenic tracers.

We are happy to make appropriate changes to the Introduction as part of framing the competing hypotheses more clearly and the role of ocean surface / atmosphere equilibrium.

The revised text has been provided in our response to the reviewer's major comments above.

L51-52: I do not understand the conclusion that atmospheric CO2 is not affected. If atmospheric CO2 (and d13C) varies/is perturbed, e.g., by outgassing in the Southern Ocean, then the CO2 perturbation will like the d13C perturbation enter the upper ocean.

The relevant text will be changed along the lines of:

"The 'top down' and 'bottom up' scenarios have different implications: In the 'bottom up' transport scenario, $\delta^{13}C$ anomaly in the marine planktic and upper intermediate depth benthic records can be used as evidence of enhanced flux of $^{13}C$-depleted carbon from the deep ocean, while in the 'top down' scenario, such an inference is invalid."

L57: This is a somewhat odd description of the preformed component. The preformed component reflects the balance between all tracer sources and sinks in a surface grid cell. Upwelling and exchange with the deeper layers are generally equally important as air-sea exchange and export production. Why highlighting the terms thermodynamic equilibrium and primary productivity? Would it not be more appropriate to mention air-sea gas exchange and new or export production as well as physical tracer exchange between surface and deeper layers?

This part will be removed (and as per our reply above).

L109: "The atmosphere is held constant ..allowed to evolve freely" This text is unclear. Do you mean in the experiment "fix" atm. CO2 and d13C is kept constant and in exp. "free" CO2 and d13C evolve freely?

Yes exactly. We will better clarify this in revision.

References:

Bernardello, R., Marinov, I., Palter, J. B., Sarmiento, J. L., Galbraith, E. D. and Slater, R. D.: Response of the Ocean Natural Carbon Storage to Projected Twenty-First-Century Climate Change, Journal of Climate, 27(5), 2033–2053, https://doi.org/10.1175/JCLI-D-13-00343.1, 2014.

Ito, T., Follows, M. J. and Boyle, E. A.: Is AOU a good measure of respiration in the oceans?: AOU AND RESPIRATION, Geophysical Research Letters, 31(17),https://doi.org/10.1029/2004GL020900, 2004.

Khatiwala, S., Schmittner, A. and Muglia, J.: Air-sea disequilibrium enhances ocean carbon storage during glacial periods, Science Advances, 5(6), eaaw4981, https://doi.org/10.1126/sciadv.aaw4981, 2019.

Menviel, L., Mouchet, A., Meissner, K. J., Joos, F. and England, M. H.: Impact of oceanic circulation changes on atmospheric $\delta^{13}CO_2$: $\delta^{13}CO_2$, Global Biogeochemical Cycles, 29(11), 1944–1961, https://doi.org/10.1002/2015GB005207, 2015.

Peterson, C. D., Lisiecki, L. E. and Stern, J. V.: Deglacial whole-ocean $\delta^{13}C$ change estimated from 480 benthic foraminiferal records, Paleoceanography, 29(6), 549–563, https://doi.org/10.1002/2013PA002552, 2014.

Schmittner, A., Gruber, N., Mix, A. C., Key, R. M., Tagliabue, A. and Westberry, T. K.: carbon isotope ratios (δ13C) in the ocean , 24, 2013.

Tschumi, T., Joos, F., Gehlen, M. and Heinze, C.: Deep ocean ventilation, carbon isotopes, marine sedimentation and the deglacial CO2 rise, Climate of the Past, 7(3), 771–800, https://doi.org/10.5194/cp-7-771-2011, 2011.

---

## Author Comment (AC3) · 9 Feb 2021

We thank the editor for handling our manuscript. We have provided detailed responses to address comments raised by both reviewers.

---

## Author Response (AR2)

I thank the authors for revising the manuscript. This manuscript has now developed into a nice paper that illustrates forcefully and convincingly the important imprint of air-sea gas exchange on upper ocean d13C.

It is nice to see that the mass balance of 13C is now treated correctly.

I have a few remaining comments that the authors may wish to consider.

We thank the reviewer for the appraisal of our work! We also appreciate the insightful suggestions made by the reviewer.

Main manuscript (line numbering refers to track changed MS)

1) l. 14 and other places: suggest to write: "a ~35ppm rise in atmospheric CO2" instead of "..pCO2" as ppm units are for a mixing ratio and not a partial pressure.

We have corrected the relevant text throughout the revised manuscript.

2) Line 487: "Since there is no 13C fractionation during CaCO3 formation in the LOVECLIM model, the last term on the RHS can be assumed to be zero (see Supplement)."

This holds for steady. The isotopic signature will change during the experiment and this change is carried downward by the CaCO3 flux (irrespective of the assumed fractionation). As described in the SI, the contribution of CaCO3 dissolution to DIC is small in the upper 1000 m and therefore the last term can also be neglected in transient experiments and the upper ocean. The authors may provide a more complete explanation.

Thanks for pointing this out. The relevant text has been changed to "Since the contribution of $CaCO_3$ dissolution is small in the upper 1000m (where GeoB17402-2 is located) in carbon cycle models (see also the Supplement), and there is no $^{13}C$ fractionation during $CaCO_3$ formation in the LOVECLIM model, the last term on the RHS can be neglected for the purpose of this study."

Supplementary Information:
1) SI: line 57: "From this we confirm, as expected, that DIC = DIC(pref) + DIC(Csoft), and d13C(DIC) = d13C(pref) + δ13C(Csoft)." I guess a qualifier is needed: "in the upper 1000 m" as CaCO3 dissolution may contribute significantly in the deep.

We admit this was not entirely clear in the text, and we intended this short paragraph to describe a simple numerical check carried out without $CaCO_3$ (and hence $\delta^{13}C_{(carb)}$) existing in the model. In the absence of any formation and dissolution of $CaCO_3$, the statement was correct. We have made this much clearer in the revised text and apologize for the ambiguity.

2) SI, line 70-90: In my opinion, it would be helpful to show here explicitly the equations how the different components are computed.
We now include equations for derived tracers #1-6 and have significantly expanded upon and hopefully further clarified all the tracer descriptions and applications.

3) Please include #7 and provide the equation describing how d13C_soft is computed from the explicitly simulated DI13C_soft and DIC_soft and DIC tracers.

We are not entirely sure what is being requested here. #5 (and newly added equation) describes how $\delta^{13}C_{(Csoft)}$ is estimated from AOU (and hence $DIC_{(Csoft)}$) together with $\delta^{13}C_{(Corg)}$ and DIC, while #6 (and newly added equation) describes how $\delta^{13}C_{(Csoft)}$ is estimated from the Csoft tracer ($DIC_{(Csoft)}$) together with $\delta^{13}C_{(Corg)}$ and DIC. The only other $\delta^{13}C_{(Csoft)}$ is the explicitly simulated numerical tracer described in an earlier section of SI.

We have revised and expanded upon the entire SI section and hope this implicitly addresses the request.

4) SI, line 93: "As is widely appreciated, AOU overestimates the consumption of oxygen through respiration as a result of incomplete equilibrium occurring between the ocean surface and overlying atmosphere." This explanation is only partly correct. It should also be mentioned in addition that the solubility of O2 is nonlinear. For example, mixing two O2 saturated water bodies with different temperatures will lead to an O2 concentration that deviates from the saturated concentration.

Thanks for pointing this out. In fact, we have greatly expanded on this discussion and also now include zonal mean fields of the AOU error itself for completeness.

Rereview of CPD manuscript 10.5194/cp-2020-95 by Shao et al.

With great interest I have reread the revised manuscript of Shao et al. and their response to my comments on the first submitted draft of the manuscript. The authors have responded efficiently and satisfactorily to many of my earlier points of criticism. The use of the LOVECLIM simulation LH1-SHW-SO is much better motivated, the logic behind a comparison of LOVECLIM and cGENIE output data is much clearer, and the paper benefits from the additional error analysis (although those should be part of the discussion rather than just a bunch of figures in the supplement). I am further glad to see that new simulations were performed with cGENIE and that the paper now focusses on the initial deglacial d13CO2 decline, which makes it clearer and more streamlined.

Thanks for the supportive comments. We also appreciate the insightful suggestions made by the reviewer.

However, on some aspects, as outlined below, the authors made, in my view, an insufficient effort to improve the impact and clarity of the study. I also noticed one other weakness, namely in the results section that includes a number of discussion elements and does not give a full account of the model results and observations that are relevant for the study.

Encouraged by the reviewer, we have expanded the results section such that the description of d13C anomaly as well as its decomposition for all ocean basins (see the new Figure3-5) are now presented for both models.

The referred 'discussion elements' previously in the result section have also been moved to the discussion section. See our response to the comments below.

Earlier I have raised concerns regarding the premise of the paper, the distinction between preformed and regenerated ocean carbon in an early deglacial transient simulation with LOVECLIM, when this model version does not simulate those species explicitly. This has been echoed by the second reviewer, who even recommended to rerun the simulations with an explicit tracer scheme for preformed and regenerated carbon. I accept the authors notion of going forward with the paper setup as is,

We appreciate that the reviewer agrees with our approach.

but I recommend to specify in line 175-177 the restrictions of the AOU approach. This is important and acknowledges that the authors approach for carbon partitioning of the LOVECLIM data is imperfect. I suggest to clearly outline the reasons why AOU (as difference between in-situ and calculated oxygen) can in some instances not be a faithful representation of regenerated carbon, and may overestimate it, as discussed in the mentioned literature.

Agree, we have changed the text to "The AOU approach to estimate respired carbon content assumes that the oxygen content of surface waters always reach equilibrium with the overlying atmosphere. However, studies have shown that this is not the case, particularly for water masses formed in high latitudes (Bernardello et al., 2014; Ito et al., 2004; Khatiwala et al., 2019, Cliff et al., 2021). As a result, AOU likely overestimates respired carbon content in the deep ocean. Additional errors associated with the AOU approach may result from the non-linear solubility of $O_2$ and respiration that does not involve $O_2$ consumption (i.e. through denitrification or sulphate reduction) (Shiller, 1981; Ito et al., 2004)."

We have also greatly expanded on this discussion in the SI and also now include zonal mean fields of the AOU error itself for completeness.

I am somewhat disappointed by the authors response to my suggestion to better carve out the relative contributions of top-down and bottom-up processes on oceanic d13C, because of the fact that this "may very much depend on the models used". The authors would have two models at hand to discuss this.

In this study, the 'top-down' and 'bottom-up' refer to two potential pathways of light d13C transport in the upper ocean. In the expanded result section, we now describe the $\Delta\delta^{13}C_{soft}$ and $\Delta\delta^{13}C_{pref}$ patterns in detail for both models. Both models show that $\Delta\delta^{13}C_{pref}$ dominates in the upper 1000m of the global ocean, which points to a top-down control. The models also show some $\Delta\delta^{13}C_{soft}$ and $\Delta\delta^{13}C_{pref}$ signals in the deep and abyssal ocean. However, the signals are more related to $\delta^{13}C$ changes in the source waters, water mass mixing ratios and ocean

ventilation state. As those changes are not the focus of this study, we chose to only briefly describe those features in the main text. Nonetheless, $\Delta\delta^{13}C$, $\Delta\delta^{13}C_{soft}$ and $\Delta\delta^{13}C_{pref}$ in both models are documented in the revised plots.

I cannot follow statements in the paper such as "Subsequently, air-sea exchange dominates the δ13C decline in the global upper ocean. (line 253)" when the cause of the d13C decline is outgassing in the Southern Ocean (LOVECLIM model). Should it say in the global upper ocean outside the Southern Ocean?

Yes, the relevant text has been changed to
"We show that in this scenario the isotopic signal is first transmitted to the atmosphere through strong outgassing in the Southern Ocean (Figure 6). The atmosphere then transmits the $\delta^{13}C$ signal to the rest of the global surface and subsurface ocean through air-sea gas exchange."

I am slightly confused, and was hoping for more clarification on the regions where outgassing occurs and where the water column is overprinted. In other words, the trigger for an atmospheric d13C bridge must be outgassing in parts of the ocean, where upper ocean d13C must see the regenerated d13C and DIC from below.

Exactly, the trigger for an atmospheric d13C bridge is through outgassing in the Southern Ocean in both LOVECLIM and cGENIE. And the upper Southern Ocean 'sees' the regenerated d13C and DIC from below.

In section 4.1, we now describe the strong outgassing in the Southern Ocean right before 'atmospheric d13C bridge' is introduced. We also revise the abstract so that this point is clearly conveyed to the readers.

"Here we present modeling evidence to show that rather than respired carbon from the deep ocean propagating directly to the upper ocean prior to reaching the atmosphere, the carbon would have first upwelled to the surface in the Southern Ocean where it enters the atmosphere. In this way the transmission of isotopically light carbon to the global upper ocean was analogous to the on-going ocean invasion of fossil fuel $CO_2$."

The water masses are likely overprinted by the atmosphere (essentially by their own signal), but it is not clear by how much. I find explanations around this issue confusing, e.g. in line 316-319: why would the atmospheric d13C_CO2 signal be compensated by the upwelling d13C_CO2 signal from the deep, when both should have a similar negative signature? Are we talking in the EEP about a gas exchange of water masses from the sub-surface (with a regenerated signal) that equilibrates with an already decreased atmospheric d13C_CO2 (acquired somewhere else)? Or has that sub-surface signal itself acquired a negative preformed signal through atmospheric CO2 overprints outside the EEP?

This is a great point. The negative d13C signal in the surface EEP can come from the gross gas exchange with an already decreased atmospheric d13C-CO2 and/or sub-surface waters that acquired a negative preformed signal at other parts of the global surface ocean. We clarify this point in the revision in the main text in section 4.2:

"On the other hand, the EEP thermocline is also shallow enough to record an atmospheric $\delta^{13}C$ signal, either directly through gas exchange at the surface or indirectly through a preformed signal acquired from other parts of the global surface ocean."

In general, I believe the paper would benefit from a more careful and in-depth explanation of the processes at play in the different ocean region, with model data support in the form of informative figures/maps. The authors mention the study of Martinez-Boti et al., (2015) which show an pCO2 oversaturation with respect to the atmospheric of 80 ppm at the beginning of the early deglaciation. This is evidence for the fact that the water mass has not fully equilibrated with the atmosphere. How can this be reconciled with the authors notion that isotopically it has acquired its signal from the atmosphere?

The isotopic equilibrium between the surface ocean and the atmosphere is achieved through gross gas exchange rather than net gas exchange. Therefore, even if the surface pCO2 in the EEP is not in equilibrium with the atmosphere in either the LOVECLIM simulation (now showed in the new Figure S9)or in the proxy reconstruction that the reviewer is referring to, it does not necessarily imply that d13C of DIC in surface water has not acquired a signal from the atmosphere.

I further recommend that the results section is not mixed with discussion elements. At its current stage, the entire Results section below line 236 (to 260) is actually discussion.
We have now moved those elements to the discussion section as the reviewer suggested

The only observation on carbon partitioning that is described and that would qualify for the results section is that from the North Pacific, which is incomplete. Even more worrisome is that none of the results of the cGENIE benchmark test has been described properly in the results section. This needs to be revised. Model papers have the luxury of presenting a wealth of data. The reader will appreciate the distillation of the main observations of the model output data.

As mentioned above, the description of the model outputs in the result section is now expanded to cover all ocean basins.

Minor comments regarding language and clarity of the text.
Line 39-40: I'd suggest to acknowledge the existing large body of literature on this. Despite extensive research efforts over the last decades, the chain of events leading to the atmospheric changes recorded in ice cores remains incompletely understood.
We have added the acknowledgement of previous research effort as suggested.
Line 57: influencing
changed as suggested.
Line 62: move "until recently (Lynch-Stieglitz et al., 2019)" to the end, otherwise this sentence makes no sense

Moved as suggested.
Line 88: I am confused by the word "theoretical". I believe that the model calculates oxygen saturation directly, as function of ocean temperature and salinity etc. Also, "at every grid point in

the model" can be dropped.

Thanks for the suggestion. The sentence now reads "Because the LOVECLIM transient experiment does not explicitly simulate either preformed or respired carbon as additional numerical tracers, the respired carbon is instead estimated by apparent oxygen utilization (AOU) – the difference between oxygen saturation and simulated $[O_2]$ (see section 2.4)."

Line 92: it is better to write "have used the separation of DIC species into regenerated and preformed as starting point/basis to study [...]" because Khatiwala et al. for instance use a much more sophisticated framework also considering disequilibrium effects.

We agree that Khatiwala et al., 2019 is not a suitable citation in this context, now the sentence reads

"The carbon partitioning framework is not new - previous studies have used this framework to study the mechanisms that lead to lower glacial atmospheric $CO_2$ (Ito and Follows, 2005; Ödalen et al., 2018; Khatiwala et al., 2019) and processes that control $\delta^{13}CO_2$ and marine carbon isotope composition (Menviel et al., 2015; Schmittner et al., 2013)."

Line 98-101: this sentence needs to be revised. E.g.: Here, we use the Earth system model of intermediate complexity cGENIE with a comprehensive diagnostic tracer framework (including for the first time a respired organic matter δ13C tracer) in order to fully evaluate the AOU-based off-line approach made based on LOVECLIM model data.

This sentence has been changed to "However, new here is the application of a 2nd Earth System model (cGENIE (Cao et al., 2009)) to fully evaluate the AOU-based off-line approach against an explicit respired organic matter $\delta^{13}C$ tracer."

Line 117: Define abbreviation VPDB
We added the definition as suggested.

Line 140: first mention of NADW, acronym needs to be defined

We added the definition as suggested.

Line 151: AOU has been defined earlier.
The repetitive definition has been removed.

Line 167: This may seem trivial, but in my view it should be mentioned how d13C_pref and DIC_pref was calculated.
We have revised the description of the cGENIE model and tracer framework in the mode text and moreover, greatly expanded and now describe in much more detail, the tracer framework in the SI.

Line 179-181: This goes against the statement in the introduction whereby the cGENIE model is employed to analyse a simulation with numerical tracer scheme.

The conflict is not clear to us. In carrying out a thorough revision of the text, we hope this has now been resolved.

Line 185: in the supplement.
We've changed the text as suggested

Lane 207: it should read Rae et al., (2020). Does that mean that the model was tuned to Pacific LGM data only? If yes, this should be explicitly stated.

Thanks for catching the citation issue; we have made the correction.

We should also clarify that the LGM configuration we use is not tuned to the North Pacific LGM data; all the boundary conditions we applied are already stated in the text. There are two reasons for not tuning cGENIE to the North Pacific LGM data in this study: 1) in our experiments, the perturbations are mainly applied to the Southern Ocean; 2) the LOVECLIM LGM state was not tuned to the North Pacific LGM data either.

Line 275: Why should this signal be seen first in the STGSP? Some explanation is needed here. Can you clarify what nature the d13C anomaly has, of regenerated or of preformed nature?

We are sorry for not being clear. The text has been changed to
"On the other hand, if the 'bottom up' scenario is true, a large negative $\delta^{13}C$ anomaly (of respired nature) should first appear in the South Pacific subtropical gyre (STGSP), as STGSP lies on the pathway between Southern Ocean water masses and those at lower latitudes."

Line 303: Reference to figure is needed. Or in fact a description in the results section.
Reference to the relevant figure in the result section is added.

Line 317: where the $\delta^{13}CO_2$ signal [...] is / Line 319: atmospheric overprint.
Changed as suggested

---

## Author Response (AR3)

Dear editor,

Thank you for the insightful comments and for handling this manuscript. We greatly appreciate your effort. We will not list the suggested grammar or typo corrections, but we have corrected them in the revised manuscript. Below we respond to your main points.

Main Text:
line 27: I am not sure what you mean by "upper-deep" waters and in which region these upper-deep waters would be located. Be more specific

The text now reads
"The model results suggest that thermocline waters throughout the ocean as well as 500-2000m water depths were affected by this atmospheric bridge during the early deglaciation."

paragraph line 205: You say you used idealized glacial boundary conditions, but then you apply interglacial CO2 of 278 ppm. This is confusing. See also comment on the supplement below.

We are sorry for not being clear. Glacial $CO_2$ concentration is not included as part of the 'idealized glacial boundary conditions' in either Rae et al., (2020) or this study. We now clearly specify the 'idealized glacial boundary conditions' applied to cGENIE. The text now reads

"For this, we take a model configuration based on the idealized 'glacial' boundary conditions of Rae et al., (2020) (including increased zonal planetary albedo at high Northern Hemisphere latitudes and the orbital configuration at 21 ka). Note, we did not attempt to achieve a glacial-like atmospheric $CO_2$ value for this spin-up, instead, we prescribed atmospheric $CO_2$ = 278ppm, $\delta^{13}CO_2$ = -6.5‰. The spin-up was run for 10,000 years.

line 284: Here you say that that the Southern Ocean would be heavier by 0.1-0.2permille everywhere, but that is not true for the surface. Please be more specific

The text now reads
"In all sectors of the Southern Ocean below 400m depth, $\delta^{13}C$ increases by 0.1-0.2‰ due to stronger ventilation."

paragraph line 405: This needs some clarification. I read it that although the EEP thermocline waters are supersaturated with respect to atmospheric CO2, their d13C is controlled by the top-down signal. This refers also to the question by referee #2 related to the work by Martinez-Boti, showing a supersaturation of waters with respect to CO2 in the atmosphere. The argument is that isotopic equilibration doesn't require net gas exchange and you should say so in this paragraph to explain this "conundrum".

Thanks for the suggestion, we added the explanation to this "conundrum". The text now reads "The modeling evidence indicates that even though the EEP is the largest $CO_2$ outgassing regions (in terms of absolute $\Delta pCO_2$, Figure S9) under an enhanced Southern Ocean upwelling scenario, its thermocline $\delta^{13}C$ is dominantly controlled by the 'top down' mechanism rather than the 'bottom up' mechanism as previously suggested (Martínez-Botí et al.,2015; Spero and Lea,

2002). The apparent conundrum can be explained by the fact that the air-sea balance of carbon isotopes is achieved through *gross* rather than *net* $CO_2$ exchange."

lines 424-425: Here you mention an early decline in d13C between 18.3 and 17 ky and that LOVECLIM would be able to simulate this. However, reading the text I wasn't sure I could pinpoint always to which process at what time you refer to. Please be more specific with the timing and clarify the text.

We are sorry for not being clear. We hope the revised text addresses the request.

"However, mid-depth (1800-2100m) benthic $\delta^{13}C$ records from the Brazil margin (~27°S) document a sharp decline of 0.4‰ at ~18 ka (Lund et al., 2019), while atmospheric $\delta^{13}CO_2$ did not decrease until ~17 ka (Bauska et al., 2016; Schmitt et al., 2012). Lund et al., (2019) argued that the lagging atmospheric $\delta^{13}CO_2$ decline seemed at odds with the idea that $\delta^{13}C_{pref}$ contributed to the early benthic $\delta^{13}C$ decrease at their site. The observed benthic $\delta^{13}C$ trend between 20-15 ka at these Brazil margin sites is well simulated by LOVECLIM (Figure 10), allowing us to explore this question further. Before atmospheric $\delta^{13}CO_2$ starts to decline in LOVECLIM at ~17.2 ka, changes in $\delta^{13}C_{DIC}$ at ~2000m depth at the Brazil Margin are dominantly controlled by excess accumulation of respired carbon (indicated by highly negative $\Delta\delta^{13}C_{soft}$, Figure S10b), itself a response to the weakened AMOC, while $\Delta\delta^{13}C_{pref}$ is relatively small (Figure S10c)."

Supplement:
line 4: Here you say that the model has only a EMBM but in the next point you say it has a dynamical sea ice model. I assume, you also have a prescribed (average) wind field? If yes, say so to clear up the contradiction

We are sorry for not being clear. Yes, the wind field is prescribed. We have added more details through the text below.

"The absence of a dynamical atmospheric GCM component then requires that (fixed, annual average) 2D fields of wind stress and speed are applied, which are re-gridded from observations, plus a zonally-average profile of planetary albedo is applied. Greenhouse gas feedback on climate is implemented by applying a top of the atmosphere anomaly in radiative forcing according to the relative deviation of atmospheric $CO_2$ from a reference value of 278 ppm. These three individual components, their coupling, plus details of the simplified atmospheric component and associated climate feedbacks, are described in *Marsh et al.* [2011] (and references therein). "

paragraph line 211: Here you explain that you do not really do a deglaciation run, but that you are only interested in the temporal evolution in a perturbed transient run. In the main text this does not become so clear, and I wonder whether you could mention that more prominently in the main text.

We were not mean to say the deglacial experiment presented in the main text is unnecessary here. We are sorry for the confusion. We hope the revised text clarify this point.

"However, although the warming approximately corresponds in overall magnitude to that associated with deglaciation and is additionally associated with reorganization of the Atlantic Meridional Overturning Circulation, it should also be noted that this idealized instantaneous-perturbation transient experiment is distinct from the deglacial-like experiment described and analyzed in the main text."